# Presynaptic PTPσ regulates postsynaptic NMDA receptor function through direct adhesion-independent mechanisms

Kyungdeok Kim[1†], Wangyong Shin[1,2†], Muwon Kang[1], Suho Lee[2], Doyoun Kim[2], Ryeonghwa Kang[1], Yewon Jung[1], Yisul Cho[3], Esther Yang[4], Hyun Kim[4], Yong Chul Bae[3], Eunjoon Kim[1,2]*

[1]Department of Biological Sciences, KAIST, Daejeon, Republic of Korea; [2]Center for Synaptic Brain Dysfunctions, Institute for Basic Science (IBS), Daejeon, Republic of Korea; [3]Department of Anatomy and Neurobiology, School of Dentistry, Kyungpook National University, Daegu, Republic of Korea; [4]Department of Anatomy and Division of Brain Korea 21, Biomedical Science, College of Medicine, Korea University, Seoul, Republic of Korea

**Abstract** Synaptic adhesion molecules regulate synapse development and function. However, whether and how presynaptic adhesion molecules regulate postsynaptic NMDAR function remains largely unclear. Presynaptic LAR family receptor tyrosine phosphatases (LAR-RPTPs) regulate synapse development through mechanisms that include trans-synaptic adhesion; however, whether they regulate postsynaptic receptor functions remains unknown. Here we report that presynaptic PTPσ, a LAR-RPTP, enhances postsynaptic NMDA receptor (NMDAR) currents and NMDAR-dependent synaptic plasticity in the hippocampus. This regulation does not involve trans-synaptic adhesions of PTPσ, suggesting that the cytoplasmic domains of PTPσ, known to have tyrosine phosphatase activity and mediate protein-protein interactions, are important. In line with this, phosphotyrosine levels of presynaptic proteins, including neurexin-1, are strongly increased in PTPσ-mutant mice. Behaviorally, PTPσ-dependent NMDAR regulation is important for social and reward-related novelty recognition. These results suggest that presynaptic PTPσ regulates postsynaptic NMDAR function through trans-synaptic and direct adhesion-independent mechanisms and novelty recognition in social and reward contexts.

*For correspondence:
kime@kaist.ac.kr

†These authors contributed equally to this work

## Introduction

Synaptic adhesion molecules regulate synapse development and function through mechanisms that include trans-synaptic adhesions in the synaptic cleft and protein interactions with cytoplasmic and membrane proteins (*de Wit and Ghosh, 2016*; *Südhof, 2017*; *Südhof, 2018*; *Shen and Scheiffele, 2010*; *Siddiqui and Craig, 2011*; *Südhof, 2018*; *Yuzaki, 2018*). Postsynaptic receptors such as NMDA and AMPA receptors (NMDARs and AMPARs) constitute an important group of synaptic proteins regulated by synaptic adhesion molecules. For instance, the synaptic adhesion molecules neuroligin-1 (*Budreck et al., 2013*) and EphB2 (*Dalva et al., 2000*) regulate the synaptic localization and function of NMDARs. In addition, neuroligin-1 acts through two distinct mechanisms to regulate NMDARs and long-term potentiation (LTP) (*Wu et al., 2019*). By binding to neuroligins, presynaptic neurexins also trans-synaptically regulate NMDARs and AMPARs. For example, presynaptic neurexin-3 promotes the postsynaptic localization of AMPARs by suppressing AMPAR endocytosis (*Aoto et al., 2013*). Alternative splicing of different neurexins also distinctly regulates postsynaptic NMDAR- and AMPAR-mediated synaptic transmission (*Dai et al., 2019*).

LAR receptor tyrosine phosphatases (LAR-RPTPs) are a family of synaptic organizers with three known members (LAR, PTPσ, and PTPδ), each with a hybrid structure that includes a single-trans-membrane domain, extracellular adhesion domains and intracellular domains D1 and D2 that possess protein tyrosine phosphatase activity and mediate protein–protein interactions, respectively (*Takahashi and Craig, 2013*; *Um and Ko, 2013*). Presynaptic LAR-RPTPs interact with postsynaptic adhesion molecules, including NGL-3 (netrin-G ligand-3, also known as leucine-rich repeat-containing 4B [LRRC4B]) (*Kwon et al., 2010*; *Woo et al., 2009*), SALM3/5 (synaptic adhesion-like molecule 3/5, also known as leucine-rich repeat and fibronectin type III domain containing 4 [LRFN4]) (*Choi et al., 2016*; *Li et al., 2015*), TrkC (tropomyosin receptor kinase C, also known as neurotrophic tyrosine kinase receptor type 3 [NTRK3]) (*Takahashi et al., 2011*), and Slitrks (SLIT and NTRK like) (*Takahashi et al., 2012*; *Yim et al., 2013*). In addition, LAR-RPTPs interact with cytoplasmic scaffolding/adaptor proteins, including liprin-α, Trio and caskin, to organize presynaptic protein complexes and promote functional differentiation (*Bomkamp et al., 2019*; *Takahashi and Craig, 2013*; *Um and Ko, 2013*). In vivo studies on LAR-RPTP–mutant mice have revealed that PTPσ regulates neurodevelopment, as evidenced by the smaller brain, sensory-motor deficits and neuroendocrine defects in these mutant mice (*Elchebly et al., 1999*; *Wallace et al., 1999*); it also regulates learning and memory (*Horn et al., 2012*). Despite this accumulating evidence of LAR-RPTP function, whether presynaptic LAR-RPTPs trans-synaptically regulate postsynaptic receptor functions, similar to neurexins, remains unknown.

In the present study, we found that presynaptic PTPσ trans-synaptically regulates the postsynaptic localization and function of NMDARs in the hippocampus. Surprisingly, this regulation does not involve trans-synaptic adhesion of PTPσ, suggesting that the cytoplasmic domains of PTPσ, possessing tyrosine phosphatase activity and mediating protein–protein interactions, are important. In line with this, a proteomic analysis revealed strong increases in phosphotyrosine (pTyr) levels in presynaptic proteins, including neurexins. Behaviorally, this trans-synaptic regulation is critical for novelty recognition in multiple assays.

## Results

### Largely normal neurodevelopmental phenotypes in *Emx1-Cre;Ptprs^{fl/fl}* mice

Previously studied PTPσ-null (*Ptprs^{−/−}*) mice display strong neurodevelopmental deficits, including neonatal death, retarded postnatal growth, and neurological and neuroendocrine deficits (*Elchebly et al., 1999*; *Wallace et al., 1999*). To circumvent these extreme phenotypes and allow more in-depth study of PTPσ function, we developed conditional knockout (cKO) mice in which a PTPσ deletion was restricted to excitatory neurons in the cortex and hippocampus by crossing *Ptprs^{fl/fl}* mice (exon four floxed) with *Emx1-Cre* mice (*Gorski et al., 2002*; *Figure 1—figure supplement 1A*). The resulting *Emx1-Cre;Ptprs^{fl/fl}* mice were genotyped by PCR (*Figure 1—figure supplement 1B*). Reductions in PTPσ protein levels in PTPσ-mutant mice were confirmed by immunoblot analysis of hippocampal samples (*Figure 1—figure supplement 1C*).

*Emx1-Cre;Ptprs^{fl/fl}* mice showed largely normal postnatal growth and survival, with a nearly normal Mendelian ratio of ~0.22 (versus the expected 0.25) and postnatal body weights (*Figure 1—figure supplement 1D*). In contrast, PTPσ global KO mice (*Ptprs^{−/−}* mice), generated in the present study, showed a strongly reduced Mendelian ratio (~0.15) and decreased body weight (~65% of WT at postnatal day [P] 21). Unlike *Ptprs^{−/−}* mice, which exhibited gait abnormalities, *Emx1-Cre;Ptprs^{fl/fl}* mice showed normal walking patterns (*Figure 1—figure supplement 1E*).

The gross morphology of the brain of *Emx1-Cre;Ptprs^{fl/fl}* mice was normal, as revealed by staining with the nuclear marker DAPI (4′,6-diamidino-2-phenylindole) (*Figure 1—figure supplement 1F*). The distribution pattern of PTPσ in the brain, revealed by X-gal staining of PTPσ-mutant mice carrying the β-Geo cassette (see *Figure 1—figure supplement 1A* for details), indicated widespread PTPσ distribution in various brain regions, including the cortex, hippocampus, striatum, thalamus, and amygdala (*Figure 1—figure supplement 1G*). These results suggest that excitatory neuron-restricted deletion of PTPσ, unlike global KO, minimally affects neurodevelopmental phenotypes.

## Normal spontaneous and basal excitatory synaptic transmission in the *Emx1-Cre;Ptprs*<sup>fl/fl</sup> hippocampus

Because previous in vitro results showed that presynaptic PTPσ regulates synapse development by interacting with multiple postsynaptic adhesion molecules (*Choi et al., 2016*; *Kwon et al., 2010*; *Li et al., 2015*; *Takahashi et al., 2011*; *Takahashi et al., 2012*; *Woo et al., 2009*; *Yim et al., 2013*), we first measured spontaneous transmission in the *Emx1-Cre;Ptprs*<sup>fl/fl</sup> hippocampus, a brain region with strong PTPσ expression (*Figure 1—figure supplement 1G*).

The frequency and amplitude of miniature excitatory postsynaptic currents (mEPSCs) and miniature inhibitory postsynaptic currents (mIPSCs) were normal in CA1 pyramidal neurons from *Emx1-Cre;Ptprs*<sup>fl/fl</sup> mice (*Figure 1A,B*).

Evoked EPSCs at Schaffer collateral-CA1 pyramidal cell (SC-CA1) synapses in *Emx1-Cre;Ptprs*<sup>fl/fl</sup> mice were also comparable to those at wild-type (WT) synapses (*Figure 1C*). In addition, paired-pulse facilitation, a measure of presynaptic function, was normal at mutant SC-CA1 synapses (*Figure 1D*).

These results suggest that excitatory neuron-restricted deletion of PTPσ does not affect spontaneous or evoked basal excitatory synaptic transmission at SC-CA1 synapses in mice. These results differ from previous results obtained using global PTPσ-null mice, which exhibit increased mEPSC frequency and increased paired-pulse facilitation at SC-CA1 synapses (*Horn et al., 2012*).

## Decreased NMDAR-dependent synaptic transmission and plasticity at *Emx1-Cre;Ptprs*<sup>fl/fl</sup> hippocampal synapses

Because synaptic organizers often regulate synaptic plasticity in addition to synapse development (*Südhof, 2018*), and LTP, but not LTD (long-term depression), is suppressed in global PTPσ-KO mice (*Horn et al., 2012*), we measured NMDAR-dependent synaptic plasticity at SC-CA1 hippocampal synapses in *Emx1-Cre;Ptprs*<sup>fl/fl</sup> mice.

LTP induced by high-frequency stimulation (HFS-LTP) was suppressed at *Emx1-Cre;Ptprs*<sup>fl/fl</sup> SC-CA1 synapses compared with that at WT synapses (*Figure 1E*). Similarly, LTP induced by theta-burst stimulation (TBS-LTP) was suppressed at these synapses (*Figure 1F*).

Intriguingly, LTD induced by low-frequency stimulation (LFS-LTD) was also suppressed at *Emx1-Cre;Ptprs*<sup>fl/fl</sup> SC-CA1 synapses (*Figure 1G*), a result that contrasts with a previous report that LFS-LTD at these synapses is normal in global PTPσ-KO mice (*Horn et al., 2012*). In contrast, metabotropic glutamate receptor (mGluR)-dependent LTD induced by the group I mGluR agonist DHPG was normal at mutant synapses (*Figure 1H*).

Because the concomitant decrease in HFS/TBS-LTP and LFS-LTD could involve decreased NMDAR-mediated synaptic transmission (*Bliss and Collingridge, 1993*; *Malenka and Bear, 2004*), we next tested if NMDAR-mediated synaptic currents were suppressed. Indeed, the ratio of NMDAR-EPSCs and AMPAR-EPSCs (NMDA/AMPA ratio) was decreased at *Emx1-Cre;Ptprs*<sup>fl/fl</sup> SC-CA1 synapses (*Figure 1I*). This result, together with the normal levels of basal excitatory synaptic transmission and mEPSCs, mediated by AMPARs, suggest that NMDAR currents are selectively decreased.

The decay kinetics of mutant NMDAR EPSCs strongly suggest that the decrease in NMDAR currents is mediated by the GluN2B subunit of NMDARs (*Figure 1I*). Indeed, levels of the GluN2B subunit were most strongly decreased in crude synaptosomal (P2) and PSD fractions, but not in the total lysates, of the *Emx1-Cre;Ptprs*<sup>fl/fl</sup> hippocampus (*Figure 1J*). The GluN2A subunit also showed a trend toward a decrease, but this difference did not reach statistical significance.

The reduced synaptic plasticity (LTP and LTD) in *Emx1-Cre;Ptprs*<sup>fl/fl</sup> mice may be attributable to reduced NMDAR-mediated synaptic transmission or changes in the signaling pathways downstream of NMDAR activation. To test this possibility, we activated NMDARs in the *Emx1-Cre;Ptprs*<sup>fl/fl</sup> hippocampus using D-cycloserine (20 μM), a glycine-site NMDAR agonist. D-cycloserine fully rescued the NMDA/AMPA ratio and TBS-LTP at SC-CA1 synapses in *Emx1-Cre;Ptprs*<sup>fl/fl</sup> hippocampal slices without affecting the NMDA/AMPA ratio or TBS-LTP at WT SC-CA1 synapses (*Figure 1K,L*).

These results collectively suggest that excitatory neuron-restricted deletion of PTPσ leads to decreases in NMDAR-mediated synaptic transmission and NMDAR-dependent synaptic plasticity, without affecting AMPAR-mediated transmission, in the hippocampal CA1 region. In addition, considering the extents of the decreases in HFS-LTP, TBS-LTP, and LFS-LTD (~44%,~66%, and ~53%,

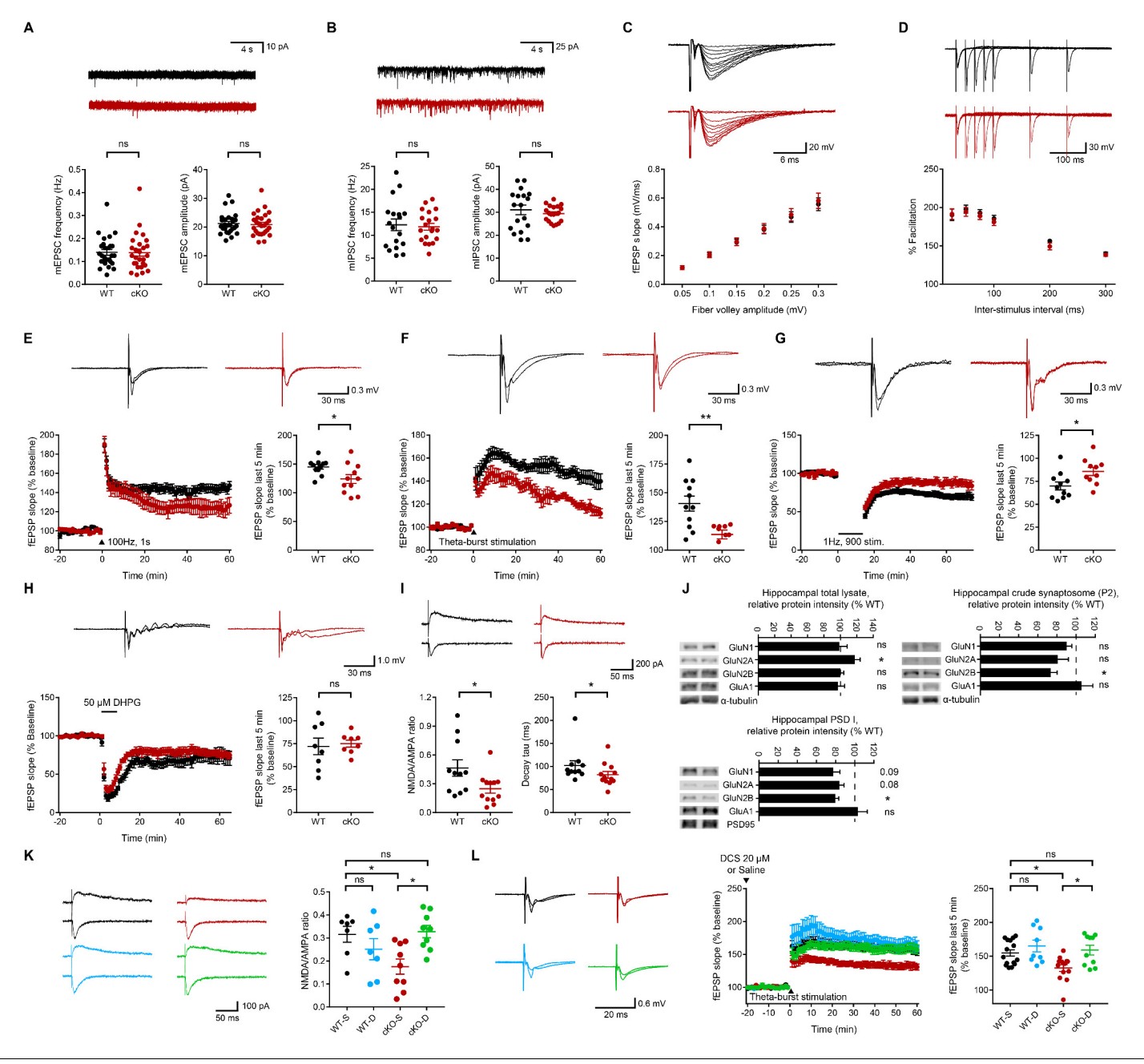

**Figure 1.** Normal spontaneous and evoked basal synaptic transmission but suppressed NMDAR-dependent synaptic transmission and plasticity in the *Emx1-Cre;Ptprs^fl/fl^* hippocampus. (**A**) Normal mEPSCs in CA1 pyramidal neurons from *Emx1-Cre;Ptprs^fl/fl^* mice (P18–22). (n = 27 cells from six mice [WT] and 28, 7 [cKO], ns, not significant, Mann-Whitney test). (**B**) Normal mIPSCs in CA1 pyramidal neurons from *Emx1-Cre;Ptprs^fl/fl^* mice (P18–22). (n = 18, 4 [WT] and 20,4 [cKO], ns, not significant,Student's t-test [amplitude], Welch's correction [frequency]). (**C**) Normal evoked basal excitatory transmission at Schaffer collateral-CA1 pyramidal cell (SC-CA1) synapses in *Emx1-Cre;Ptprs^fl/fl^* mice (P26-30), as shown by fEPSP slopes plotted against fiber volley amplitude. (n = 14 slices from 5 mice and 14, 6 [WT, cKO], ns, not significant, repeated measures two-way ANOVA). (**D**) Normal paired-pulse ratio at SC-CA1 synapses of *Emx1-Cre;Ptprs^fl/fl^* mice (P26–30), as shown by percent facilitation plotted against inter-pulse intervals. (n = 17, 5 [WT] and 15, 5 [cKO], ns, not significant, repeated-measures/RM two-way ANOVA). (**E**) Suppressed HFS-LTP at *Emx1-Cre;Ptprs^fl/fl^* SC-CA1 synapses (P26-32). (n = 11, 6 [WT] and 11, 6 [cKO], *p<0.05, Student's t-test). (**F**) Suppressed TBS-LTP at *Emx1-Cre;Ptprs^fl/fl^* SC-CA1 synapses (P26-32). (n = 11, 4 [WT] and 9, 4 [cKO], **p<0.01, Student's t-test). (**G**) Suppressed LFS-LTD at *Emx1-Cre;Ptprs^fl/fl^* SC-CA1 synapses (P16-19). (n = 11, 6 [WT] and 10, 5 [cKO], *p<0.05, Student's t-test). (**H**) Normal mGluR-LTD induced by DHPG (50 μM) at *Emx1-Cre;Ptprs^fl/fl^* SC-CA1 synapses (2–3 weeks). (n = 8, 7 [WT] and 8, 6 [cKO], ns, not significant, Student's t-test). (**I**) Decreases in the ratio of NMDAR-EPSCs and AMPAR-EPSCs and the decay tau of NMDAR-EPSCs at *Emx1-Cre;Ptprs^fl/fl^* SC-CA1 synapses (P18-23). (n = 11 cells from five mice [WT] and 12, 5 [cKO], *p<0.05, Student's t-test [NMDA/AMPA ratio], Mann-Whitney test [decay tau]). (**J**) Decreased levels of the GluN2B, but not GluN1 or GluN2A, subunit of NMDARs in crude synaptosomal (**P2**) and PSD I fractions, but not in

*Figure 1 continued on next page*

*Figure 1 continued*

total lysates, of the *Emx1-Cre;Ptprs*$^{fl/fl}$ hippocampus (3 weeks), compared with those in WT mice. α-tubulin was blotted for controls. GluA1, AMPAR subunit. (n = 7 mice (WT/cKO total lysates, 6,4 [WT and cKO P2], n = 3, 3 [WT and cKO PSD], *p<0.05, one sample t-test). (K) D-cycloserine (20 μM) rescues the NMDA/AMPA ratio at *Emx1-Cre;Ptprs*$^{fl/fl}$ SC-CA1 synapses (P19–22). (n = 7 cells from four mice [WT-V/vehicle], 7, 3 [WT-D/D-cycloserine], 9, 4 [cKO-V], 9, 4 [cKO-D], *p<0.05, ns, not significant, two-way ANOVA with Sidak's test). (L) D-cycloserine (20 μM) rescues TBS-LTP at at *Emx1-Cre;Ptprs*$^{fl/fl}$ SC-CA1 synapses (P28–32), without affecting TBS-LTP at WT synapses. (n = 15 slices from six mice [WT-V], 13, 5 [cKO-V], 9, 3 [WT-D], 10, 3 [cKO-D], *p<0.05, ns, not significant, two-way ANOVA with Sidak's test).

The online version of this article includes the following figure supplement(s) for figure 1:

**Figure supplement 1.** Generation and basic characterization of global *Ptprs*-mutant mice and *Emx1-Cre;Ptprs*$^{fl/fl}$ mice.

---

respectively) and the decrease in the NMDA/AMPA ratio (~45%) at the mutant synapses under naïve and D-cycloserine rescue conditions (*Figure 1E–I and K,L*), the decreased LTP and LTD seem to mainly involve decreased NMDAR currents rather than signaling pathways downstream of NMDAR activation. In addition, the decreased levels of GluN2B in the PSD fraction (~20%) may contribute partly to the decrease in NMDAR currents (~45%).

## Presynaptic, but not postsynaptic, deletion of PTPσ suppresses hippocampal LTP

Because presynaptic PTPσ trans-synaptically interacts with postsynaptic adhesion molecules (e.g., NGL-3, TrkC, Slitrks, and IL1RAPL1) (*Choi et al., 2016*; *Kwon et al., 2010*; *Li et al., 2015*; *Takahashi et al., 2011*; *Takahashi et al., 2012*; *Woo et al., 2009*; *Yim et al., 2013*), we tested the possibility that presynaptic loss of PTPσ suppresses postsynaptic LTP.

To this end, we deleted PTPσ in presynaptic neurons in the hippocampal CA3 region and measured LTP at SC-CA1 synapses in the CA1 area by injecting AAV1-hSyn-Cre-eGFP into the CA3 region of *Ptprs*$^{fl/fl}$ mice ~ 2.5 weeks prior to LTP measurement at ~4 weeks (*Figure 2A*). In control experiments, we injected AAV1-hSyn-Cre-eGFP into the CA1 (postsynaptic) area and measured LTP at SC-CA1 synapses. Specific expression of Cre recombinase in the CA3 or CA1 region was confirmed by monitoring EGFP (enhanced green fluorescent protein) signals. Reduced levels of PTPσ protein (~20–40% of WT) were confirmed by immunoblot analyses of hippocampal samples from Cre-expressing CA3 and CA1 areas (*Figure 2B*).

LTP experiments indicated that Cre-induced deletion of PTPσ in the CA3 region suppresses TBS-LTP at SC-CA1 synapses in *Ptprs*$^{fl/fl}$ mice compared with control synapses expressing EGFP alone (no Cre) (*Figure 2C*). In contrast, deletion of PTPσ in the CA1 region had no effect on TBS-LTP. These results suggest that PTPσ in the presynaptic (CA3) region, but not the postsynaptic (CA1) region, is important for normal LTP at SC-CA1 synapses.

## Re-expression of presynaptic PTPσ rescues postsynaptic LTP in the hippocampus

To further test the hypothesis that presynaptic PTPσ regulates postsynaptic LTP, we re-expressed PTPσ in presynaptic CA3 neurons by locally injecting AAV-eIF1a-Ptprs into the CA3 region of *Emx1-Cre;Ptprs*$^{fl/fl}$ mice (*Figure 3A*).

In addition to WT PTPσ constructs, we used constructs of mutant PTPσ with extracellular mutations that abrogate trans-synaptic interactions with postsynaptic/extracellular adhesion molecules (TrkC, Slitrk1, HSPG, and CSPG) (*Figure 3B,C*; *Figure 3—figure supplement 1*). Expression levels and molecular weights of these PTPσ mutants were verified by immunoblot analysis of HEK293T cell lysates using two different PTPσ antibodies (targeting N- and C-termini) (*Figure 3D*).

In control experiments, in which control virus without PTPσ (AAV-hSyn-eGFP) was injected into the CA3 region of *Emx1-Cre;Ptprs*$^{fl/fl}$ or *Ptprs*$^{fl/fl}$ (control) mice, TBS-LTP at *Emx1-Cre;Ptprs*$^{fl/fl}$ synapses was smaller than that at *Ptprs*$^{fl/fl}$ (control) synapses (*Figure 3E*), recapitulating TBS-LTP results from naïve (un-injected) mice (*Figure 1F*).

Re-expression of WT PTPσ in the CA3 region of *Emx1-Cre;Ptprs*$^{fl/fl}$ mice by local injection of AAV-eIF1a-Ptprs rescued TBS-LTP at SC-CA1 synapses, restoring it to levels comparable to those in *Ptprs*$^{fl/fl}$ (control) mice injected with control virus (AAV-hSyn-eGFP) (*Figure 3E*). These results

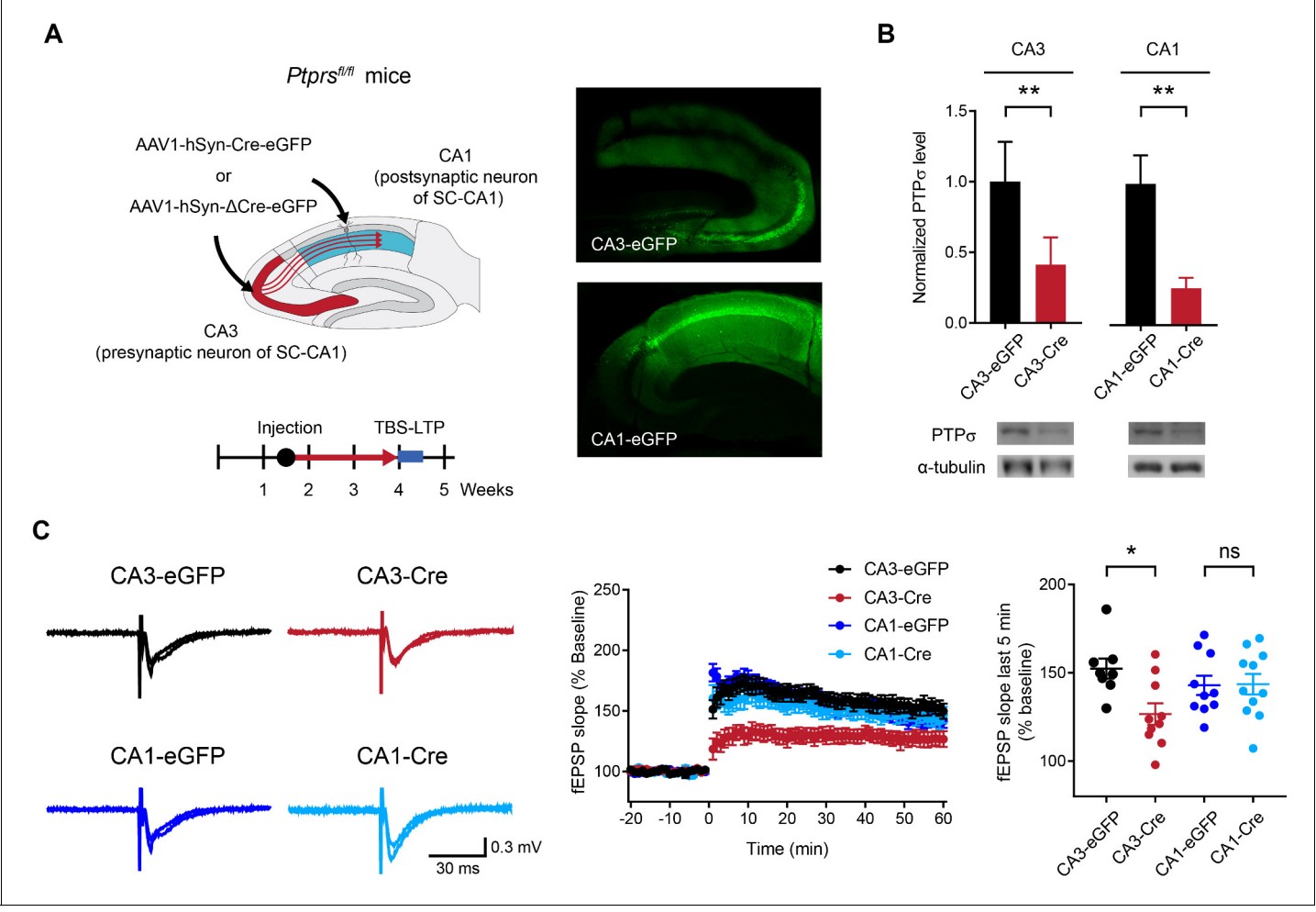

**Figure 2.** Deletion of presynaptic, but not postsynaptic, PTPσ suppresses TBS-LTP at SC-CA1 synapses. (**A**) Diagram depicting deletion of presynaptic PTPσ in the CA3 region or postsynaptic PTPσ in the CA1 region, induced by local injection of AAV1-hSyn-Cre-eGFP (Cre recombinase fused to EGFP); mice injected with AAV1-hSyn-ΔCre-eGFP (lacking Cre) served as controls. TBS-LTP experiments were conducted ~2.5 weeks later. Local expression of Cre recombinase was further confirmed by EGFP fluorescence in controls co-injected with AAV1-hSyn-eGFP (right) and by reduced PTPσ protein levels (panel B). (**B**) Decreased PTPσ protein levels in CA3 and CA1 regions of *Ptprs^fl/fl* mice injected with AAV1-hSyn-Cre-eGFP (CA3/CA1-Cre) compared with those in *Ptprs^fl/fl* mice injected with AAV1-hSyn-ΔCre-eGFP (CA3/CA1-EGFP). (n = 5 mice [CA3-eGFP], 5 [CA3-Cre], 4 [CA1-eGFP] and 3 [CA1-Cre], *p<0.05, **p<0.01, one-sample t-test). (**C**) Suppressed TBS-LTP at SC-CA1 synapses of *Ptprs^fl/fl* mice (4 weeks) induced by knocking out PTPσ in the CA3 region (but not CA1 region) by local injection of AAV1-hSyn-Cre-eGFP, compared with TBS-LTP in control *Ptprs^fl/fl* mice injected with AAV-hSyn-ΔCre-eGFP. (n = 8 slices from five mice [CA3-eGFP], 10, 4 [CA3-Cre], 10, 4 [CA1-eGFP] and 11, 5 [CA1-Cre], *p<0.05, ns, not significant, two-way ANOVA with Sidak's test).

suggest that acute presynaptic re-expression of PTPσ in CA3 rescues postsynaptic LTP in the CA1 region.

## Extracellular regions of PTPσ are not important for postsynaptic LTP regulation

We next tested whether trans-synaptic adhesions of PTPσ are important for postsynaptic LTP regulation using mutant PTPσ proteins that lack HSPG/CSPG or Slitrk1/TrkC interactions (K68A/K69A/K71A/K72A and Y224S, respectively) (see *Supplementary file 1* for details) (*Coles et al., 2014*; *Coles et al., 2011*; *Han et al., 2018*; *Um et al., 2014*; *Won et al., 2017*). Surprisingly, re-expression of either PTPσ mutant in the CA3 region by local AAV injection rescued TBS-LTP to an extent similar to that of WT PTPσ injection (*Figure 3E*), suggesting that HSPG/CSPG and Slitrk1 interactions are not important for the rescue of LTP.

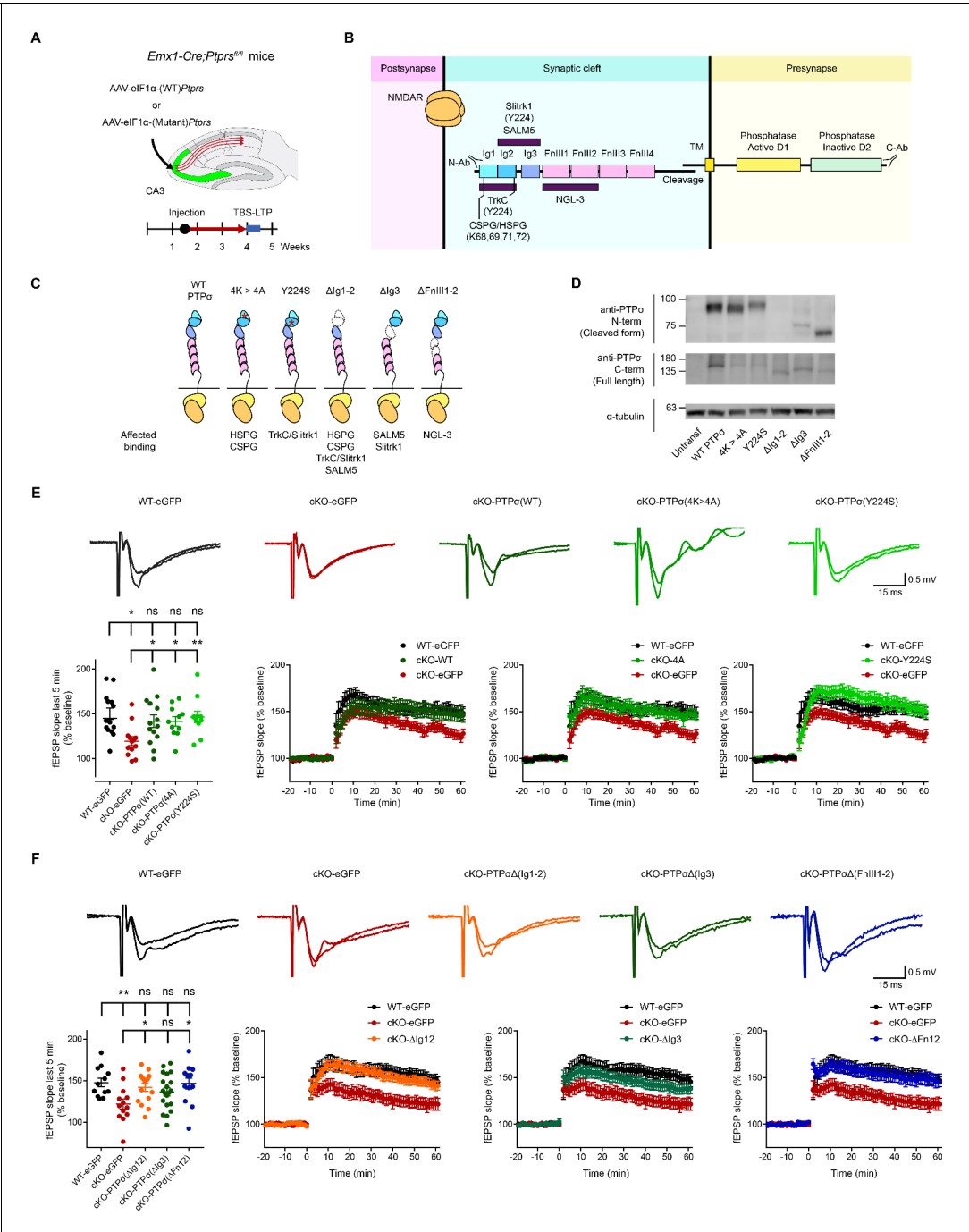

**Figure 3.** Re-expression of presynaptic PTP σ rescues postsynaptic LTP in the hippocampus through mechanisms independent of the extracellular region of PTPσ. (A) Diagram depicting re-expression of WT and mutant PTPσ proteins in the CA3 region of the hippocampus by local injection of AAV (php.eB)-eIF1a-Ptprs (WT and mutants), followed by measurement of TBS-LTP. (B) Diagram depicting the domain structures of PTPσ and extracellular and cytoplasmic regions/domains involved in protein-protein interactions or tyrosine phosphatase activity. The first three Ig domains are important for trans-synaptic adhesions with Slitrk1, TrkC, CSPG/HSPG or SALM5, and the residues Y224 and K68/K69/K71/K72 are important for Slitrk1 and CSPG/HSPG interactions, respectively. (C) Specific PTPσ mutants used in our experiments, with point mutations or small deletions in the extracellular domains. All binding partners of PTPσ affected by the mutations/deletions are also indicated. (D) Expression levels and sizes of PTPσ mutants, revealed by immunoblot analysis of HEK293T cell lysates using two independent PTPσ antibodies targeting the N-terminal region (~Ig1-2) and C-terminus (last 30 residues) that can detect all PTPσ mutants, except for PTPσ-ΔIg12, which is not detected by the N-terminal antibody. (E) Rescue of TBS-LTP at SC-CA1 synapses by re-expression of WT PTPσ as well as mutant PTPσ (lacking CSPG/HSPG and Slitrk1 interactions) in the CA3 region of *Emx1-Cre;Ptprs^fl/fl* mice (P28–32) through local injection of AAV-eIF1a-Ptprs-WT/mut. In control experiments, AAV-hSyn-eGFP was injected into the CA3 region of both *Ptprs^fl/fl* (WT) and *Emx1-Cre;Ptprs^fl/fl* mice. (n = 14 slices from four mice [WT-eGFP], 13, 5 [cKO-eGFP], 14, 6 [cKO-PTPσ-WT], 11, 4 [cKO-PTPσ−4A], and

*Figure 3 continued on next page*

Figure 3 continued

11, 4 [cKO-PTPσ-Y224S], *p<0.05, **p<0.01, ns, not significant, one-way ANOVA with Dunnett's test). (F) Full and partial rescue of TBS-LTP at SC-CA1 synapses by re-expression of mutant PTPσ lacking Ig1+2 (ΔIg12), Ig3 (ΔIg3), or FNIII1+2 (ΔFN12) domains in the CA3 region of *Emx1-Cre;Ptprs*^fl/fl^ mice (P28–32) by local injection of AAV-eEF1-Ptprs. Note that expression of PTPσ-ΔIg3 induces a partial rescue. Control virus (AAV-eIF1a-eGFP) was injected into the CA3 region of both *Ptprs*^fl/fl^ (control) and *Emx1-Cre;Ptprs*^fl/fl^ mice. (n = 12 slices from four mice [WT-eGFP], 13, 4 [cKO-eGFP], 16, 5 [cKO-PTPσ-ΔIg12], 20, 6 [cKO-PTPσ-ΔIg3], and 13, 5 [cKO-PTPσ-ΔFn12], *p<0.05, **p<0.01, ns, not significant, one-way ANOVA with Dunnett's test).

The online version of this article includes the following figure supplement(s) for figure 3:

**Figure supplement 1.** PTPσ adhesions with known postsynaptic partners, and identification of the amino acid residues involved in the interactions.

However, these PTPσ mutants might not cover as yet unknown trans-synaptic or extracellular binding partners of PTPσ and their contribution to postsynaptic LTP regulation. We thus generated PTPσ mutants carrying small deletions of Ig1+2, Ig3, or FNIII1-2 domains (*Figure 3C*) (*Supplementary file 1*). However, all of these PTPσ mutants rescued TBS-LTP at SC-CA1 synapses in the *Emx1-Cre;Ptprs*^fl/fl^ hippocampus when re-expressed in the CA3 region, although the PTPσ mutant containing an Ig3 deletion produced only partial rescue (*Figure 3F*). These results collectively suggest that the extracellular domains or regions of PTPσ are not important for PTPσ-dependent postsynaptic regulation of LTP.

## Synaptic proteins with altered pTyr levels in the *Emx1-Cre;Ptprs*^fl/fl^ cortex and hippocampus

Presynaptic PTPσ-dependent regulation of postsynaptic LTP does not involve extracellular regions of PTPσ, suggesting that the cytoplasmic region of PTPσ is important. This region contains the D1 and D2 domains, which are known to possess tyrosine phosphatase activity and mediate interactions with cytoplasmic proteins, respectively, with the latter (D2 domain) potentially linking the D1 domain with its pTyr substrates. We thus employed a proteomic approach to perform an unbiased search of proteins with altered pTyr levels in the *Emx1-Cre;Ptprs*^fl/fl^ cortex and hippocampus using anti-pTyr antibodies followed by liquid chromatography tandem mass spectrometry (LC-MS/MS) to pull down and identify pTyr proteins (*Figure 4A*).

This proteomic analysis revealed that, of 1549 proteins (3894 pTyr motifs), 57 proteins (80 pTyr motifs) showed significant changes in pTyr levels in mutant mice compared with WT mice (p<0.05; fold change >1.5), as indicated by a volcano plot (*Figure 4B*; *Supplementary file 2*). These significant changes included both upregulation (29 proteins/33 motifs) and downregulation (29 proteins/47 motifs) of pTyr levels. Among the proteins with significantly changed pTyr levels in mutant mice was LRRTM3, which showed both up and downregulation of phosphorylation at different pTyr motifs.

The identified proteins were mainly receptors/channels, adaptors/scaffolds, protein kinases, small GTPase regulators, and adhesion/extracellular matrix/cell surface proteins (*Supplementary file 2*). Specific examples include GluN2A and GluN2B (receptors/channels); PSD-95/Dlg4, GKAP/SAPAP4 and Shank3 (adaptor/scaffold); EphA4/A5/B1, PKCα/β, TrkC and Lyn (protein kinases); SynGAP1 (small GTPase regulators); and neurexin-1, Neto1, APP/amyloid-beta A4 protein, vGluT2/Slc17a6, synaptotagmin-11 and δ2-catenin (adhesion/extracellular matrix/cell surface proteins).

Proteins of particular interest among those that were differentially tyrosine phosphorylated include the GluN2B subunit of NMDARs with altered phosphorylation at tyrosine residues 985, 997, 1004, 1367 and 1369. Given the decreased P2 and PSD, but not total, levels of GluN2B and decreased NMDAR currents in the *Emx1-Cre;Ptprs*^fl/fl^ hippocampus (*Figure 1J*), these changes in pTyr, which have not been previously reported, suggest that altered phosphorylation of GluN2B may regulate the synaptic localization or function of GluN2B. Another protein of interest was TrkC, a postsynaptic adhesion partner of PTPσ (*Takahashi et al., 2011*) that was found to be differentially phosphorylated at tyrosine residues 597 and 604. Although these pTyr residues have not been studied previously, these findings suggest that maintenance of normal tyrosine phosphorylation of TrkC at these residues is dependent on presynaptic PTPσ. Other proteins of interest included neurexins, which regulate NMDAR- and AMPAR-mediated synaptic responses (*Dai et al., 2019*); Neto1, a trans-membrane protein that regulates synaptic localization of NMDARs and kainate receptors, and regulates LTP and learning and memory (*Cousins et al., 2013*; *Molnár, 2013*; *Ng et al., 2009*;

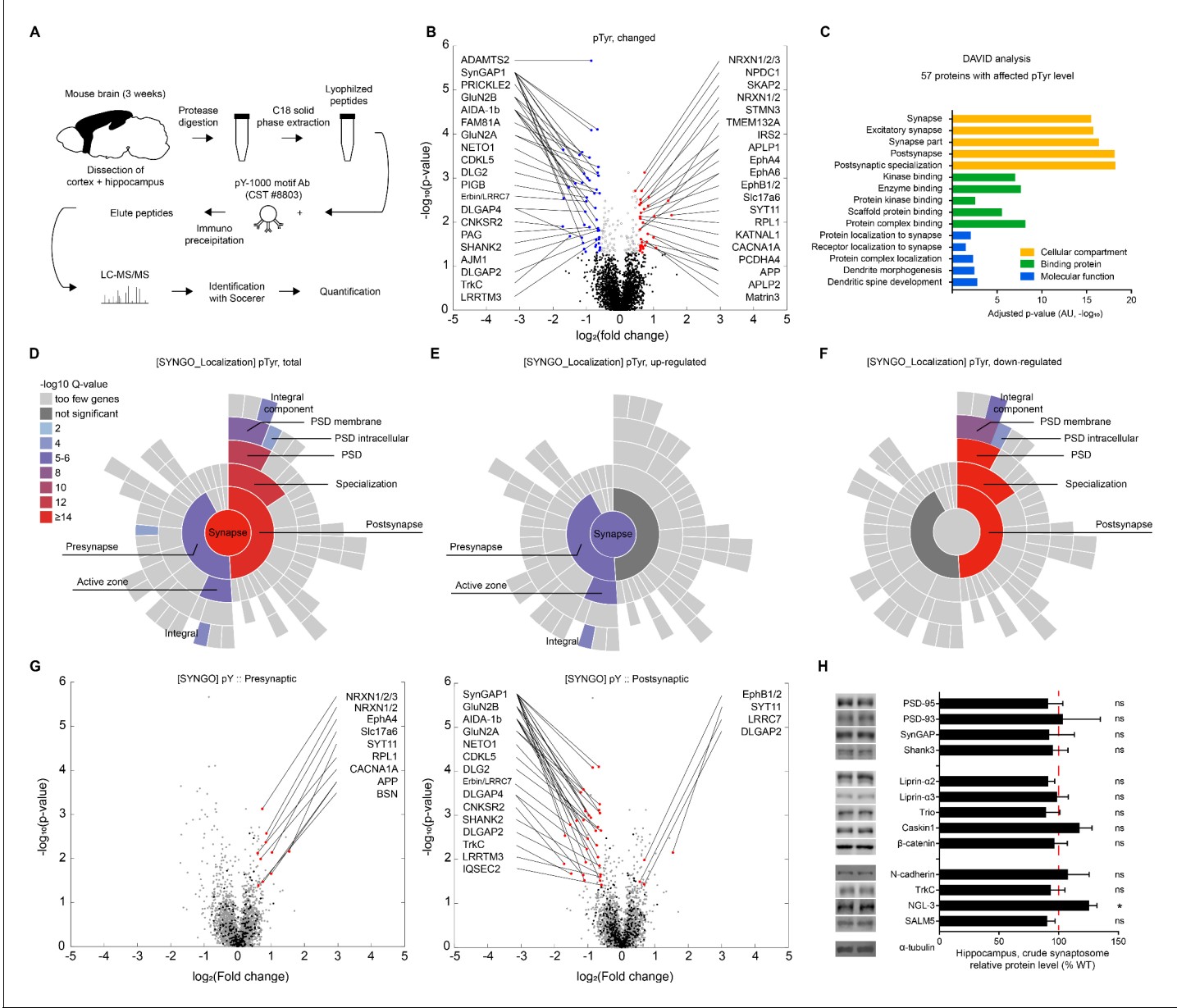

**Figure 4.** Proteins with altered pTyr levels in the *Emx1-Cre;Ptprs<sup>fl/fl</sup>* cortex and hippocampus. (**A**) Diagram showing procedures for the identification of proteins with altered pTyr levels. (**B**) A volcano plot showing proteins with significantly up- or downregulated pTyr levels (fold change >1.5, p<0.05) in the cortex and hippocampus of *Emx1-Cre;Ptprs<sup>fl/fl</sup>* (P21). Samples from three WT or KO mice were pooled for each analysis. Note that a single protein can be linked to multiple dots (different pTyr motifs); conversely, the same peptide can belong to two different proteins (e.g., Erbin and LRRC7). (**C**) Functional analysis (DAVID GO analysis) of proteins from *Emx1-Cre;Ptprs<sup>fl/fl</sup>* mice with significantly altered pTyr levels. Note that synapse-related GO terms are strongly enriched. (**D–F**) Sunburst plots from SynGO analyses of specific pre- and postsynaptic localizations of proteins with altered pTyr levels (total, upregulated, and downregulated) from the *Emx1-Cre;Ptprs<sup>fl/fl</sup>* cortex and hippocampus. Note that upregulated proteins tend to be those that localize to presynaptic sites, whereas downregulated proteins tend to be those that localize to postsynaptic sites. (**G**) Volcano plots showing that presynaptic proteins, determined by SynGO analyses, showed mainly increased pTyr levels, whereas postsynaptic proteins showed mainly decreased pTyr levels. (**H**) Largely normal synaptic levels of PTPσ-related proteins in the hippocampus of *Emx1-Cre;Ptprs<sup>fl/fl</sup>* mice (3 weeks), as shown by immunoblot analyses of crude synaptosomes. (n = 6 mice for WT and cKO and for some, 6, five mice for WT and cKO, *p<0.05, ns, not significant, one-sample t-test).

The online version of this article includes the following figure supplement(s) for figure 4:

**Figure supplement 1.** SynGO analysis of the functions of proteins with altered pTyr Levels in the *Emx1-Cre;Ptprs<sup>fl/fl</sup>* cortex and hippocampus.

*Wyeth et al., 2014*); and APP, which associates with and regulates the trafficking of NMDARs (*Cousins et al., 2013*; *Snyder et al., 2005*).

A DAVID Gene Ontology (GO) analysis (http:// david.ncifcrf.gov) of proteins with significant pTyr levels showed that GO terms in the cellular compartment module with the strongest scores were synapse related, and included 'synapse', 'excitatory synapse' and 'postsynaptic specialization' (*Figure 4C*). Other strong GO terms included 'kinase binding' and 'scaffold protein binding', in the protein binding module, and 'protein/receptor localization to synapse' and 'dendritic spine development', in the molecular function module. Therefore, proteins with altered pTyr levels were those that are strongly associated with synapse organization and protein-protein interactions.

Application of SynGO analysis, a recently reported set of expert-curated GO terms for synaptic proteins (https://www.syngoportal.org/; *Koopmans et al., 2019*), to proteins with altered pTyr levels from *Emx1-Cre;Ptprs^{fl/fl}* mice showed that 29 of the 57 proteins with significant pTyr changes corresponded to synaptic proteins in the SynGO database. These proteins fell into diverse functional categories; notably, proteins exhibiting upregulated pTyr levels were linked to 'synapse organization' and 'presynapse' functions, whereas those with downregulated pTyr levels were linked to 'synapse organization' and 'postsynapse' functions (*Figure 4—figure supplement 1B,C*), indicative of distinct pre- and postsynaptic functions depending on up- versus downregulation of pTyr, respectively.

Additional SynGO analyses of protein localization showed that these pTyr proteins were localized to both pre- and postsynaptic sites (*Figure 4D*). Intriguingly, upregulated proteins were more enriched at presynaptic than postsynaptic sites (9 presynaptic and four postsynaptic) (*Figure 4E*), whereas downregulated proteins were more strongly enriched at postsynaptic than presynaptic sites (15 postsynaptic and 0 presynaptic) (*Figure 4F*).

Volcano plot displays of these pre- and postsynaptic proteins, determined based on the SynGO analysis, further highlighted the fact that presynaptic proteins showed mainly increased pTyr levels, whereas postsynaptic proteins showed mainly decreased pTyr levels (*Figure 4G*). These results collectively suggest that excitatory neuron-restricted deletion of PTPσ leads to strong changes in pTyr levels in synaptic proteins, and primarily increases pTyr levels of presynaptic proteins and decreases pTry levels of postsynaptic proteins. These findings further indicate that upregulated presynaptic pTyr proteins might represent potential pTyr substrates of PTPσ.

Changes in pTyr levels in the abovementioned synaptic proteins may influence their synaptic function or localization. These changes may also occur in PTPσ-interacting presynaptic proteins, such as liprin-α and caskin (*Bomkamp et al., 2019*; *Serra-Pagès et al., 1998*). In addition, deletion of PTPσ may lead, directly or indirectly, to the loss of postsynaptic partners of PTPσ or the PTPσ-dependent regulation of postsynaptic plasticity.

To test these possibilities, we investigated whether synaptic levels of PTPσ-related proteins are decreased by performing immunoblot analyses of crude synaptosomes from WT and *Emx1-Cre; Ptprs^{fl/fl}* mice. We found no changes in the synaptic levels of D2 domain-interacting proteins (liprin-α 1/2 and Caskin 1/2) or known substrates of PTPσ (N-cadherin and β-catenin) (*Siu et al., 2007*; *Figure 4H*). Moreover, there were no changes in the synaptic levels of postsynaptic scaffolding proteins (PSD-95, PSD-93, SynGAP, and Shank3) or postsynaptic binding partners of PTPσ (SALM5 and NGL-3), although there was a moderate increase in NGL-3 levels. These results suggest that deletion of PTPσ has minimal impacts on the synaptic localization of PTPσ-related proteins.

## Suppressed novelty recognition in *Emx1-Cre;Ptprs^{fl/fl}* mice

Hippocampal NMDAR-dependent LTP and LTD have been linked to multiple types of learning and memory behaviors (*Bliss et al., 2003*; *Malenka and Bear, 2004*). We thus first subjected *Emx1-Cre; Ptprs^{fl/fl}* mice to novel object-recognition tests, in which a subject mouse is exposed to two identical objects on day 1, and one of the two objects is replaced with a new object on day 2. Unlike WT mice, *Emx1-Cre;Ptprs^{fl/fl}* mice failed to recognize the novel object on day 2 (*Figure 5A*). The increase in baseline object exploration (~2 folds) in the mutant mice, partly attributable to the increased locomotion and object exploration (~20% and~30%, respectively, n = 16 mice), is less likely to affect the relative exploration of familiar and novel objects.

To test if this lack of novel-object preference is specific for an object but not for a novel mouse, we next subjected mice to a three-chamber test, designed to test for social approach and social-novelty recognition (*Silverman et al., 2010*). *Emx1-Cre;Ptprs^{fl/fl}* mice showed normal social approach, as indicated by the preference for a social target (stranger mouse) over an object, but failed to show

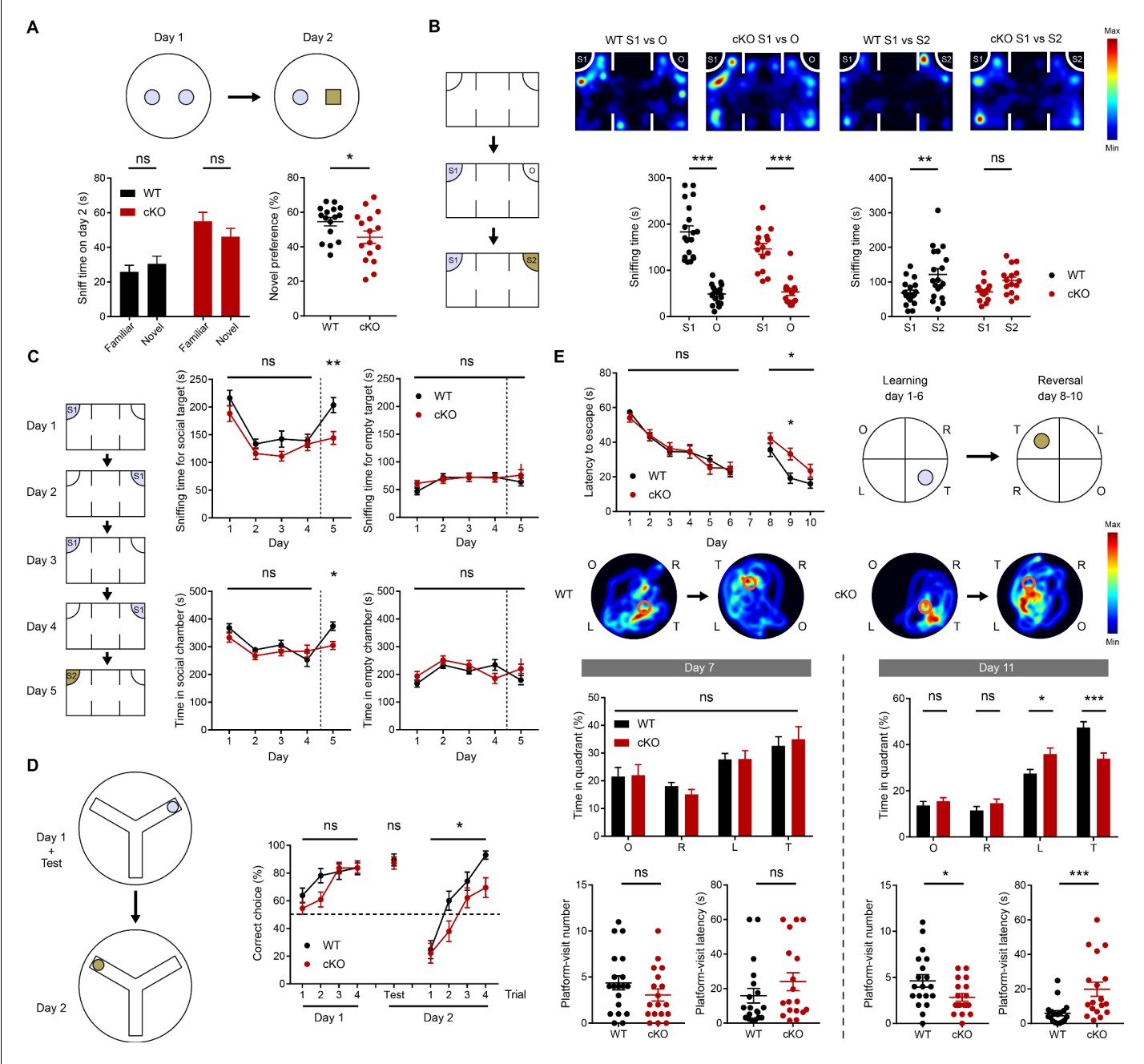

**Figure 5.** Suppressed novelty recognition in *Emx1-Cre;Ptprs*<sup>fl/fl</sup> mice. (**A**) Impaired novel-object recognition in *Emx1-Cre;Ptprs*<sup>fl/fl</sup> mice (2–3 months). (n = 16 mice for WT and cKO, *p<0.05, ns, not significant, Student's t-test). (**B**) Normal social approach, but suppressed social-novelty recognition, in *Emx1-Cre;Ptprs*<sup>fl/fl</sup> mice (2–3 months) in the three-chamber test. S1, first/initial social stranger; O, object; S2, second/new social stranger. (n = 19 [WT] and 15 [cKO], **p<0.01, ***p<0.001, ns, not significant, RM two-way ANOVA with Sidak's test). (**C**) Normal social recognition and habituation, but suppressed social-novelty recognition, in *Emx1-Cre;Ptprs*<sup>fl/fl</sup> mice (2–3 months) in a modified three-chamber test, in which the subject mouse was exposed to the initial stranger mouse for four consecutive days and introduced to a new stranger mouse on day 5. (**D**) Suppressed reward-arm recognition in the reversal, but not initial, phase of the Y-maze in *Emx1-Cre;Ptprs*<sup>fl/fl</sup> mice (2–3 months). (n = 21 [WT] and 22 mice [cKO] for initial session, n = 17, 19 for reversal session, *p<0.05, ns, not significant, RM two-way ANOVA [initial/reversal] and Mann-Whitney test [test on day 2]). (**E**) Impaired learning and memory in the reversal, but not initial, phase of the Morris water maze in *Emx1-Cre;Ptprs*<sup>fl/fl</sup> mice (3–4 months). (n = 19 [WT] and 18 [cKO], *p<0.05, ***p<0.001, ns, not significant, RM two-way ANOVA with Sidak's test [latency to escape], Student's t-test [platform visit number] and Mann-Whitney test [platform visit latency]).

The online version of this article includes the following figure supplement(s) for figure 5:

*Figure 5 continued on next page*

*Figure 5 continued*

**Figure supplement 1.** *Emx1-Cre;Ptprs*<sup>fl/fl</sup> mice show moderately changed locomotion and anxiety-like behavior, but normal motor coordination, repetitive behavior, and prepulse inhibition.

normal social novelty preference, similarly exploring new and old stranger mice (*Figure 5B*). These results suggest that *Emx1-Cre;Ptprs*<sup>fl/fl</sup> mice fail to recognize both a novel object and a novel mouse.

To further explore this phenotype, we performed a modified social interaction test in which a subject mouse was repeatedly exposed to the initial stranger mouse for four consecutive days and then was introduced to a new stranger mouse on day 5 (*Bariselli et al., 2018*). *Emx1-Cre;Ptprs*<sup>fl/fl</sup> mice spent increasingly less time with the initial stranger mouse over the first four days, indicative of normal social cognition and habituation, but spent less time with a novel mouse on day five compared with WT mice (*Figure 5C*), further confirming the decrease in social-novelty recognition.

In the Y-maze test, where the reward arm on day one was switched to another arm on day 2, *Emx1-Cre;Ptprs*<sup>fl/fl</sup> mice displayed less efficient switching to the novel arm containing the reward on day 2 (*Figure 5D*). Lastly, in the Morris water maze test, *Emx1-Cre;Ptprs*<sup>fl/fl</sup> mice showed normal learning and memory in the initial phase. However, in the reversal phase, in which the location of the hidden platform was switched to a new quadrant, the mutant mice were less efficient in switching to the novel platform location (*Figure 5E*). In other behavioral tests, *Emx1-Cre;Ptprs*<sup>fl/fl</sup> mice showed normal levels of motor coordination, repetitive behaviors, and prepulse inhibition (*Figure 5—figure supplement 1*). Notably, these mice showed moderately decreased locomotor activity in a familiar environment (Laboras cages) and moderately increased locomotor activity in a novel environment (open-field apparatus). In addition, these mice showed moderately increased open-field center time and strongly increased open-arm time in the elevated plus-maze test, suggestive of anxiolytic-like behavior, although they performed normally in the light-dark chamber test.

These results indicate that *Emx1-Cre;Ptprs*<sup>fl/fl</sup> mice display decreases in the ability to recognize a novel object (novel-object recognition test), a novel social target (three-chamber test), a novel reward-arm location (Y-maze test) and a novel platform location (Morris water maze), collectively suggesting that the mutant mice have suppressed novelty recognition.

## Presynaptic PTPσ-dependent regulation of postsynaptic NMDARs is important for novelty recognition

The impaired novel recognition in *Emx1-Cre;Ptprs*<sup>fl/fl</sup> mice may involve PTPσ-dependent regulation of postsynaptic LTP. To assess this possibility, we first tested whether pharmacological activation of NMDARs could rescue the impaired social-novelty recognition in *Emx1-Cre;Ptprs*<sup>fl/fl</sup> mice.

Acute treatment with the NMDAR agonist D-cycloserine (20 mg/kg; i.p.) rescued social-novelty recognition deficits in *Emx1-Cre;Ptprs*<sup>fl/fl</sup> mice, as assessed using the three-chamber test, without affecting social-novelty recognition in WT mice (*Figure 6A*). In addition, D-cycloserine had no effect on social approach in WT or *Emx1-Cre;Ptprs*<sup>fl/fl</sup> mice.

We next tested whether presynaptic, but not postsynaptic, deletion of PTPσ affects novelty recognition by injecting AAV1-hSyn-Cre-eGFP or control AAV1-hSyn-ΔCre-eGFP into the CA3 or CA1 region of *Ptprs*<sup>fl/fl</sup> mice (8–11 weeks). Cre-induced PTPσ deletion in the CA3 region resulted in impaired social-novelty recognition in *Ptprs*<sup>fl/fl</sup> mice in the three-chamber test compared with control *Ptprs*<sup>fl/fl</sup> mice expressing EGFP alone, without affecting social approach (*Figure 6B,C*). In contrast, Cre-induced PTPσ deletion in the CA1 region had no effect on social-novelty recognition or social approach.

Cre-induced deletion of PTPσ in the CA3 region also impaired recognition of the novel reward arm location in the Y-maze test in *Ptprs*<sup>fl/fl</sup> mice compared with control *Ptprs*<sup>fl/fl</sup> mice expressing EGFP alone (*Figure 6D*). These results suggest that deletion of presynaptic, but not postsynaptic, PTPσ impairs social novelty and novel reward-arm recognition in adult mice.

To more directly test whether NMDAR function in the postsynaptic CA1 area is important for social novelty and reward-arm recognition, we acutely knocked down expression of the GluN1 subunit of NMDARs in the CA1 region of *Ptprs*<sup>fl/fl</sup> mice (8–11 weeks) and monitored its impacts on novelty recognition. WT mice (C57BL/6J) injected in the CA1 region with AAV-pU6-shGluN1 displayed suppressed social-novelty recognition, but normal social approach, in the three-chamber test

(*Figure 6E,F*). In contrast, control WT mice injected with AAV-pU6-shCtrl displayed normal social-novelty recognition and social approach.

In addition, WT mice injected in the CA1 region with AAV-pU6-shGluN1 failed to recognize the novel reward-arm location in the Y-maze test, whereas control WT mice similarly injected with AAV-pU6-shCtrl showed no such change (*Figure 6G*). Decreased expression of GluN1 was verified by immunoblot analysis of the GluN1 protein expressed in the infected hippocampus (*Figure 6H*). Therefore, normal expression of NMDARs in the CA1 region is important for social novelty and novel reward-arm recognition.

These results collectively suggest that presynaptic PTPσ-mediated regulation of postsynaptic NMDAR currents and NMDAR-dependent LTP is important for social novelty and novel reward-arm recognition in mice.

## Discussion

Our results suggest that presynaptic PTPσ regulates postsynaptic NMDAR currents and NMDAR-dependent LTP. This conclusion is supported by several lines of evidence: 1) NMDAR-dependent synaptic transmission and plasticity are decreased in the *Emx1-Cre;Ptprs*$^{fl/fl}$ hippocampus; 2) decreased NMDAR currents and LTP at *Emx1-Cre;Ptprs*$^{fl/fl}$ SC-CA1 synapses are rescued by the NMDAR agonist D-cycloserine; 3) presynaptic (CA3), but not postsynaptic (CA1), deletion of PTPσ suppresses LTP; and 4) re-expression of PTPσ in presynaptic (CA3) neurons rescues LTP.

Our data also suggest mechanisms by which presynaptic PTPσ regulates postsynaptic NMDAR currents and LTP. Our data suggest that the decreased synaptic levels of GluN2B at mutant excitatory synapses, supported by immunoblot analysis of synaptic proteins and the faster decay kinetics of NMDAR currents, may partly contribute to the decreased NMDAR currents, which in turn seems to underlie the reduced LTP, based on quantitative comparisons. In addition, point mutations or small deletions in the extracellular region of PTPσ that disrupt trans-synaptic adhesions of Ig1-3 or FNIII1-2 domains with TrkC, SALM5, Slitrk1, CSPG/HSPG, and NGL-3 do not affect the PTPσ-dependent regulation of postsynaptic LTP, suggesting that the cytoplasmic regions of PTPσ, containing the tyrosine phosphatase activity and mediating presynaptic protein-protein interctions, may be important.

Our proteomic analysis provides potential clues to the specific mechanisms underlying PTPσ-dependent regulation of NMDAR currents and postsynaptic LTP. Deletion of PTPσ led to strong increases in pTyr levels in presynaptic proteins, but strong decreases in postsynaptic proteins. Presynaptic proteins with increased pTyr levels included neurexin-1/2/3, EphA4, vGluT2/Slc17a6, synaptotagmin-11, CACNA1A/voltage-dependent calcium channel (P/Q-type), APP, and Bassoon. Among these proteins, presynaptic neurexin-1, but not neurexins 2/3, promotes NMDAR-mediated, but not AMPAR-mediated, postsynaptic responses (*Dai et al., 2019*). Therefore, PTPσ, which is linked to neurexin-1 through liprin-α and CASK (*Hata et al., 1996*; *Olsen et al., 2005*; *Serra-Pagès et al., 1995*) or caskin (*Bomkamp et al., 2019*; *Weng et al., 2011*), may contribute to the neurexin-1–dependent regulation of postsynaptic NMDARs. The tyrosine residues in neurexin-1 with upregulated phosphorylation are found at multiple, previously unreported locations in the protein. Equally intriguing is that the other presynaptic proteins with altered pTyr levels identified in the present study (EphA4, vGluT2, synaptotagmin-11, CACNA1A, APP, and Bassoon) regulate presynaptic differentiation/function as well as postsynaptic NMDAR function, likely through indirect mechanisms. For instance, LTP can be regulated by EphA4 (*Filosa et al., 2009*), APP (*Weyer et al., 2011*), and synaptotagmin-11 (*Shimojo et al., 2019*).

The decrease in pTyr levels in major postsynaptic proteins in the *Emx1-Cre;Ptprs*$^{fl/fl}$ cortex and hippocampus suggests that deletion of PTPσ and consequent changes in presynaptic proteins/functions may affect postsynaptic proteins and their pTyr levels. Mechanisms underlying the strong postsynaptic decreases (not increases) in pTyr levels are unclear but may involve altered activities of tyrosine kinases and phosphatases at postsynaptic sites, as suggested by altered p-Tyr levels in these proteins (*Supplementary file 2*), or the decreased function of NMDARs, known to bidirectionally interact with and regulate various tyrosine kinases and phosphatases such as Src family proteins, Eph receptors, and STEP (*Henderson and Dalva, 2018*; *Sala and Sheng, 1999*; *Salter and Kalia, 2004*). Although further details remain to be determined, these results suggest the possibility that postsynaptic proteins, specifically those with decreased pTyr levels, may regulate NMDAR currents

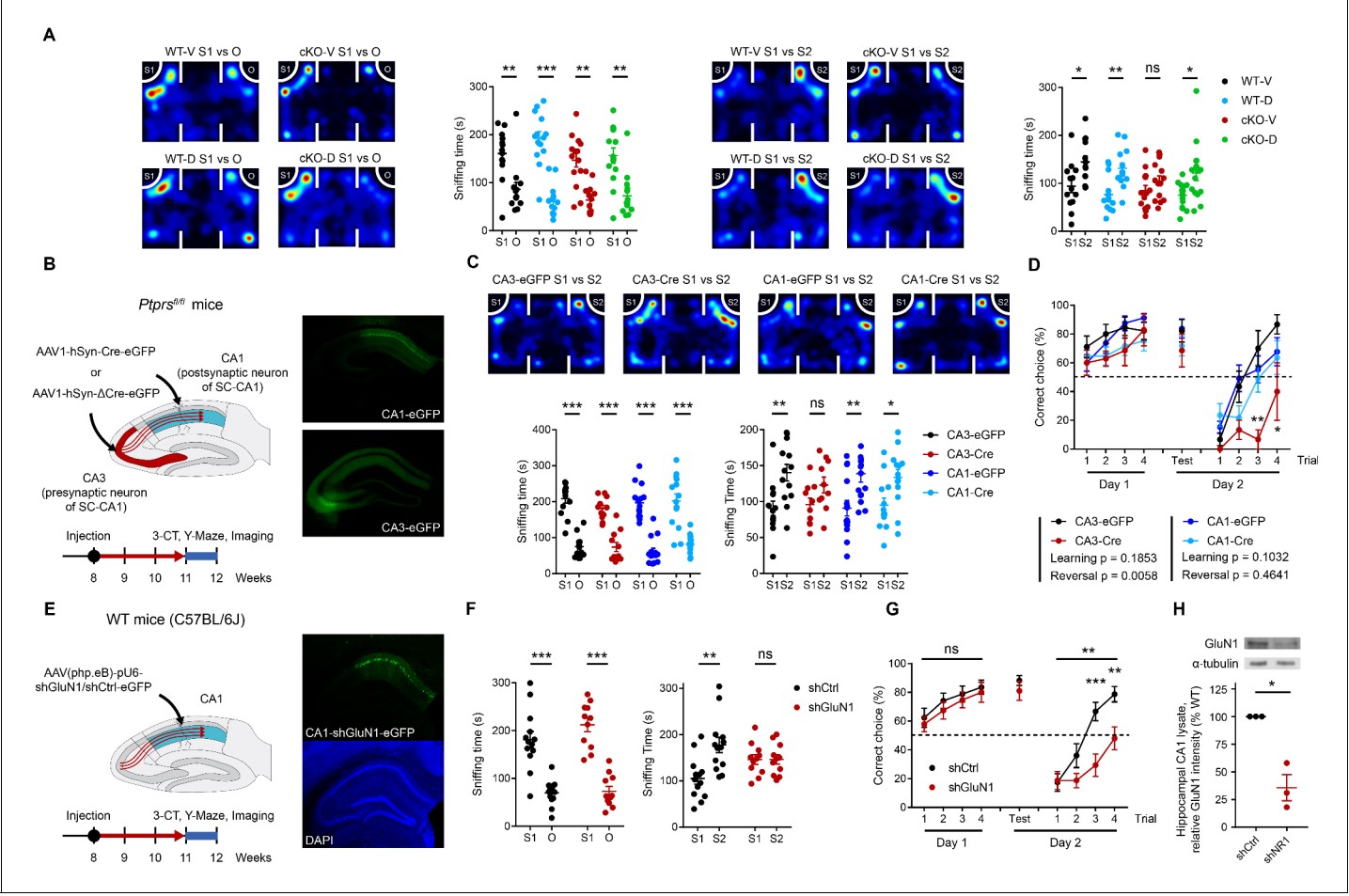

**Figure 6.** Presynaptic PTPσ-dependent regulation of postsynaptic NMDARs is important for novelty recognition in *Emx1-Cre;Ptprs^{fl/fl}* mice. (**A**) D-cycloserine (DCS) treatment (20 mg/kg; i.p.) rescues social-novelty recognition in *Emx1-Cre;Ptprs^{fl/fl}* mice (2–3 months) without affecting social approach. Note that DCS has no effect on social approach or novelty recognition in WT mice. (n = 13, 14, 14, 14 mice for WT-V/vehicle, WT-D/D-cycloserine, cKO-V, cKO-D, *p<0.05, **p<0.01, ***p<0.001, ns, not significant, two-way ANOVA with Sidak's test). (**B and C**) Impaired social-novelty recognition in *Ptprs^{fl/fl}* mice injected with AAV1-hSyn-Cre-eGFP in the CA3 region, but not the CA1 region, in the three-chamber test compared with control *Ptprs^{fl/fl}* mice injected with AAV1-hSyn-ΔCre-eGFP in CA3 or CA1 regions. Note that social approach (left panel) is unaffected by either CA3- or CA1-specific PTPσ KO. EGFP fluorescence indicates virus injection sites in CA3/CA1 regions. (n = 13, 11, 15, 14 mice for CA3-EGFP, CA3-Cre, CA1-EGFP and CA1-Cre, respectively, *p<0.05, **p<0.01, ***p<0.001, ns, not significant, two-way ANOVA with Sidak's test). (**D**) Impaired recognition of novel, but not initial, reward-arm location in *Ptprs^{fl/fl}* mice injected with AAV1-hSyn-Cre-eGFP in the CA3, but not CA1, region in the Y-maze test (3 months), compared with control *Ptprs^{fl/fl}* mice injected with AAV1-hSyn-ΔCre-eGFP. (n = 9, 7, 16, 17 mice for CA3-EGFP, CA3-Cre, CA1-EGFP and CA1-Cre, respectively, during learning phase, n = 6, 3, 13, 11 mice for reversal phase, two-way ANOVA with Sidak's test) (**E and F**) Knockdown of the GluN1 subunit of NMDARs in the CA1 region of WT mice (C57/BL6J) by injection of AAV(php.eB)-pU6-shGluN1 suppresses social-novelty recognition but not social approach, a finding that contrasts with the normal social-novelty recognition observed in control WT mice injected with AAV(php.eB)-pU6-shCtrl (scrambled control). (n = 13, 11 mice for shCtrl and shGluN1, respectively, **p<0.01, ***p<0.001, ns, not significant, two-way ANOVA with Sidak's test). (**G**) GluN1 knockdown in the CA1 region of WT mice (C57/BL6J) by injection of AAV(php.eB)-pU6-shGluN1 suppresses novel, but not initial, reward-arm recognition compared with control WT mice injected with AAV(php.eB)-pU6-shCtrl. (n = 17 [shGluN1-initial], 19 [shCtrl-initial], 15 [shGluN1-reversal], 15 [shCtrl-reversal], **p<0.01, ***p<0.001, ns, not significant, RM two-way ANOVA with Sidak's test). (**H**) Validation of AAV(php.eB)-pU6-shGluN1/shCtrl viruses by immunoblot analysis of the GluN1 protein from the injected hippocampus. GluN1 levels were normalized to α-tubulin levels. (n = 3, three mice for ShCtrl and shGluN1, *p<0.05, one sample t-test).

and LTP. Indeed, it has been reported that NMDARs or LTP are regulated by GluN2A/B (*Collingridge and Bliss, 1995*; *Malenka and Bear, 2004*), NETO1 (*Ng et al., 2009*), EphB1/2 (*Dalva et al., 2000*; *Henderson and Dalva, 2018*), PSD-93/DLG2/Chapsyin-110 (*Carlisle et al.,*

*2008*), SynGAP (*Araki et al., 2015*; *Komiyama et al., 2002*), SAPAP/DLGAP2/4 (*Schob et al., 2019*), Shank2 (*Schmeisser et al., 2012*; *Won et al., 2012*), and CDKL5 (*Okuda et al., 2017*). The fact that none of the pTyr residues in these proteins identified in the present study has been previously reported may form the basis for future studies of NMDAR or LTP regulation by these pTyr proteins.

Our data also suggest that presynaptic PTPσ regulates not only postsynaptic NMDAR currents and LTP, but also behavioral novelty recognition. This hypothesis is supported by the observations that 1) D-cycloserine rescues social novelty deficits in *Emx1-Cre;Ptprs*^fl/fl^ mice, 2) acute presynaptic (CA3) deletion of PTPσ impairs social novelty and novel reward-arm recognition, and 3) acute postsynaptic (CA1) knockdown of the GluN1 subunit of NMDARs impairs social novelty and novel reward-arm recognition. Previous studies have implicated the hippocampus and hippocampal NMDARs in the regulation of novelty recognition in various contexts (*Hitti and Siegelbaum, 2014*; *Kitanishi et al., 2015*; *Leroy et al., 2017*; *Rondi-Reig et al., 2001*). In addition, hippocampal LTD but not LTP has been associated with novel-object/feature recognition in a space (*Dong et al., 2012*; *Kemp and Manahan-Vaughan, 2004*; *Kemp and Manahan-Vaughan, 2007*). Our data, in particular those from CA3-PTPσ–KO and CA1-GluN1–knockdown experiments, extend previous findings by demonstrating that a presynaptic adhesion molecule—PTPσ—can regulate social and reward-arm novelty recognition through trans-synaptic regulation of postsynaptic NMDAR currents and NMDAR-dependent LTP in the hippocampus. Whether our results would also involve the rescue of NMDAR-dependent LTD, which is impaired in the *Emx1-Cre;Ptprs*^fl/fl^ hippocampus, remains to be determined. In addition, care should be taken in interpreting our results because *Emx1-Cre;Ptprs*^fl/fl^ mice lack *Ptprs* expression not only in the hippocampus but also in other brain regions such as the prefrontal cortex, known to be involved in social novelty cognition in mice and rats (*Cao et al., 2018*; *Finlay et al., 2015*; *Liang et al., 2018*; *Niu et al., 2018*; *Watson et al., 2012*).

Notably, *Emx1-Cre;Ptprs*^fl/fl^ mice also display anxiolytic-like behavior as supported by moderately increased center time in the open-field test and strongly increased open-arm time in the elevated plus-maze test, although these mice acted normally in the light-dark test. Whether the anxiolytic-like behavior involves suppressed NMDAR function in the hippocampus or other brain regions is an open question. Previous studies have shown that anxiety involves various brain regions, including anterior cingulate cortex, hippocampus, and amygdala (*Adhikari, 2014*; *Apps and Strata, 2015*; *Barthas et al., 2015*; *Calhoon and Tye, 2015*; *Duval et al., 2015*; *Kim et al., 2011*; *Tovote et al., 2015*).

Lastly, a recent paper has reported that deletion of all three LAR-RPTPs (PTPσ, PTPδ, and LAR) minimally affects synapse development but critically regulate postsynaptic NMDAR, but not AMPAR, responses by a trans-synaptic mechanism (*Sclip and Südhof, 2020*). These results are in line with our results that PTPσ deletion in mice selectively decreases NMDAR, but not AMPAR, currents. This study also extends our study by finding that PTPδ and LAR, in addition to PTPσ, participate in the regulation of trans-synaptic NMDAR regulation. An obvious direction for follow-up studies based on these results would be to identify specific mechanisms underlying the trans-synaptic but indirect NMDAR regulation.

In conclusion, our results suggest that presynaptic PTPσ regulates postsynaptic NMDAR currents and NMDAR-dependent LTP in the hippocampus through trans-synaptic adhesion-independent mechanisms, and suggest that this regulation may be important for novelty recognition in social- and reward-related contexts.

# Materials and methods

**Key resources table**

| Reagent type (species) or resource | Designation | Source or reference | Identifiers | Additional information |
| --- | --- | --- | --- | --- |
| Gene (*Mus musculus*) | *Ptprs* | | 19280 in ncbi | For iteration of next studies. |

*Continued on next page*

*Continued*

| Reagent type (species) or resource | Designation | Source or reference | Identifiers | Additional information |
|---|---|---|---|---|
| Strain, strain background (*Mus musculus; C57BL/6J*) | *Emx1-Cre;Ptprs*<sup>fl/fl</sup>; *Ptprs*<sup>−/−</sup> mice | ES-Cell from KOMP: Ptprs<sup>tm1a(KOMP)Mbp</sup> | RRID: MGI_5797751 | gKO/cKO mice used in this study |
| Genetic reagent (*Mus musculus*) | Emx1-Cre | JAX | #005628 | Cre-expressing line used in this study |
| Transfected virus (*Mus musculus*) | pAAV1-hSyn-Cre-eGFP | Addgene | #105540 | Experimental virus for Cre injection (*Figure 2* and *Figure 6*) |
| Transfected virus (*Mus musculus*) | pAAV1-hSyn-ΔCre-eGFP | Addgene | #105539 | Control virus for Cre injection (*Figure 2* and *Figure 6*) |
| Transfected construct (*Mus musculus*) | pAAV-nEFCas9 | Addgene | #87115 | Vector for WT/Mut Ptprs expression (*Figure 3*). |
| Transfected construct (*Mus musculus*) | Ptprs WT gene | *Li et al., 2015* | Ptprs(meA-/meB-) | Detailed sequence is added in *Supplementary file 1*. This used for mutagenesis and WT Ptprs rescue injection |
| Transfected construct (*Mus musculus*) | pAV-pU6-shGluN1#1-GFP | Vigene | #SH836303 | Experimental virus for Cre injection (*Figure 6*) |
| Transfected construct (*Mus musculus*) | pAV-pU6-shscrambled-GFP | Vigene | #SH836303 | Control virus for Cre injection (*Figure 6*) |
| Comparative phosphor-proteomic analysis | PhosphoSCAN service | Cell Signalling Technology | Phospho-tyrosine (pY-1000) | Used in *Figure 4* |
| Antibody | anti-PTPσ (Guinea Pig polyclonal) | This paper | #2135 for N-term epitope, #2138 for C-term epitope | WB(1:500) |
| Antibody | anti-PSD93/SynGAP1/NGL3/SALM5/GluA1); (Rabbit,mouse polyclonal) | Home-made; used in previous studies from our group. | #1634(PSD93); #1682(SynGAP1); #2020(NGL3)#1943(SALM50;#1193(GluA1) | WB(1:500) |
| Antibody | anti-tubulin (mouse monoclonal) | DSHB | 12G10 | WB(1:5000) |
| Antibody | anti-GluN1(mouse))/GluN2A(rabbit)) monoclonal | Millipore | Mab363(GluN1); 07-632(GluN2A) | WB(1:500) |
| Antibody | anti-GluN2B(mouse)/PSD95(mouse) monoclonal | NeuromAb | 73-101(GluN2B); 75-028(PSD95) | WB(1:500) |
| Antibody | anti-Shank3 (Rabbit;poly)/Synaptophysin (Mouse;mono) | Santa Cruz | H160(Shank3); D4(Synaptophysin) | WB(1:500) |
| Antibody | anti-linprinα3/caskin1 (Rabbit polyclonal) | Synaptic Systems | 169 102(Liprin α3); 185 003(caskin1) | WB(1:500) |
| Antibody | anti-liprinα2 (Rabbit)/Trio(Mouse) polyclonal | Abcam | Ab155411(liprinα2); 194365(Trio) | WB(1:500) |
| Antibody | anti-βcatenin (Mouse monoclonal) | BD Science | 610154 | WB(1:500) |
| Antibody | anti-N-Cadherin (Mouse monoclonal) | Thermo | 33–3900 | WB(1:500) |

*Continued on next page*

*Continued*

| Reagent type (species) or resource | Designation | Source or reference | Identifiers | Additional information |
|---|---|---|---|---|
| Antibody | anti-TrkC (Rabbit monoclonal) | Cell Signaling Technology | 3376 | WB(1:500) |
| Software, algorithm | GraphPad Prism 7.0 | GraphPad | Ver 7.0 | Used for all statistics used in the current study. |
| Software, algorithm | DAVID analysis | David.ncifcrf.gov | DAVID analysis | Used for statistics in proteomics |

## Mice

We received ES cells containing a *Ptprs*-targeted allele from KOMP (RRID:MGI:5797751; Ptprs$^{tm1a}$ $^{(KOMP)Mbp}$), and transgenic mice were generated through ES injection. We backcrossed it with C57BL/6J strains for more than five generations before we conduct experiments. After mating with *Protamine-Flp*, the resulting *Ptprs*$^{fl/+}$ mice were crossed with *Emx1-Cre* mice (JAX #005628) to produce *Emx1-Cre;Ptprs*$^{fl/fl}$ mice. For *Ptprs* global knockout mice (*Ptprs* gKO mice), we treated fertilized eggs at the two-cell embryo stage with purified HTNC, a cell-permeable Cre recombinase (Histidine-TAT-Nuclear localization-Cre fusion peptide (*Peitz et al., 2002*), in a media at the final concentration of 0.3 μM for 30–40 mins. *Emx1-Cre;Ptprs*$^{fl/fl}$ mice were genotyped by polymerase chain reaction (PCR) using the following primer sets: Ptprs allele, forward, 5'-CTCCTTCCTCTCCAAACGG-3', reverse, 5'-TGAGCGTCTGAATGGAGCAC-3', Cre allele, forward, 5'-GATCTCCGGTATTGAAAC TCCAGC-3', reverse, 5'-GCTAAACATGCTTCATCGTCGG-3'. Appropriate expression patterns of *Emx1-Cre* was confirmed by crossing with ROSA-tdTomato mice (JAX #7909). All mice were housed and bred at the mouse facility of Korea Advanced Institute of Science and Technology (KAIST) and maintained according to the Animal Research Requirements of KAIST. All animals were fed ad libitum and housed under 12 hr light/dark cycles (light phase during 1 am to one pm). We crossed *Ptprs*$^{fl/fl}$ male mice and *Emx1-Cre;Ptprs*$^{fl/fl}$ female mice to produce littermate pairs of wild-type (WT) and KO mice. Mice were weaned at the age of postnatal day 21, and mixed-genotype littermates in the same gender were housed together until experiments. All procedures were approved by the Committee of Animal Research at KAIST (KA2016-33).

## Electrophysiology

For electrophysiological experiments for the hippocampus, sagittal hippocampal slices (400 μm thickness for extracellular recordings and 300 μm for intracellular recordings) from *Emx1-Cre;Ptprs*$^{fl/fl}$ mice, or virus-injected mice, and their appropriate controls (see each figures) were prepared using a vibratome (Leica VT1200) in ice-cold dissection buffer containing (in mM) 212 sucrose, 25 NaHCO$_3$, 5 KCl, 1.25 NaH$_2$PO$_4$, 0.5 CaCl$_2$, 3.5 MgSO$_4$, 10 D-glucose, 1.25 L-ascorbic acid and 2 Na-pyruvate bubbled with 95% O$_2$/5% CO$_2$. For virus-injected samples, slices with fluorescence signals derived from co-injected AAV1-hSyn-eGFP were used. The slices were recovered at 32°C for 1 hr in normal ACSF (in mM: 125 NaCl, 2.5 KCl, 1.25 NaH$_2$PO$_4$, 25 NaHCO$_3$, 10 glucose, 2.5 CaCl$_2$ and 1.3 MgCl$_2$ oxygenated with 95% O$_2$/5% CO$_2$). For electrophysiological recordings, a single slice was moved to and maintained in a submerged-type chamber at 28°C, continuously perfused with ACSF (2 ml/min) saturated with 95% O$_2$/5% CO$_2$. Stimulation and recording pipettes were pulled from borosilicate glass capillaries (Harvard Apparatus) using a micropipette electrode puller (Narishege).

For extracellular recordings, mouse hippocampal slices at the age of postnatal days 16–33 were used (for the exact ages of each experiment, see S1_Table). fEPSPs were recorded in the stratum radiatum of the hippocampal CA1 region using pipettes filled with ACSF (1 MΩ). fEPSPs were amplified (Multiclamp 700B, Molecular Devices) and digitized (Digidata 1440A, 1550 Molecular Devices) for analyses. The Schaffer collateral pathway was stimulated every 20 s with pipettes filled with ACSF (0.3–0.5 MΩ). The stimulation intensity was adjusted to yield a half-maximal response, and three successive responses were averaged and expressed relative to the normalized baseline. To induce LTP or LTD, high-frequency stimulation (100 Hz, 1 s), theta-burst stimulation (40 trains of pulses, each train is composed with 4 stimuli in 100 Hz; 40 trains are divided by four bursts, each containing 10 trains with 1 s inter-burst interval; 170 ms inter-train-interval), or low-frequency stimulation (1 Hz, 15 min) were applied after a stable baseline was acquired. To induce mGluR dependent

LTD, DHPG (50 µM) was added to ACSF for 5 min after acquiring a stable baseline. The paired-pulse ratio was measured across a range of inter-stimulus intervals of 25, 50, 75, 100, 200, and 300 ms. For D-cycloserine (DCS) rescue, 20 µM DCS-containing ACSF were used to perfuse slices from the beginning of baseline recording.

Whole-cell patch-clamp recordings of hippocampal CA1 pyramidal neurons were made using a MultiClamp 700B amplifier (Molecular Devices) and Digidata 1440A, 1550 (Molecular Devices). During whole-cell patch-clamp recordings, series resistance was monitored each sweep by measuring the peak amplitude of the capacitance currents in response to short hyperpolarizing step pulse (5 mV, 40 ms); only cells with a change in <20% were included in the analysis. For afferent stimulation of hippocampal pyramidal neurons, the Schaffer collateral pathway was selected. For NMDA/AMPA ratio experiments, mouse hippocampal slices (P19–23) were used. Recording pipettes (2.5–3.5 MΩ) were filled with an internal solution containing the following (in mM): 100 CsMeSO$_4$, 10 TEA-Cl, 8 NaCl, 10 HEPES, 5 QX-314-Cl, 2 Mg-ATP, 0.3 Na-GTP, and 10 EGTA, with pH 7.25, 295 mOsm. CA1 pyramidal neurons were voltage clamped at −70 mV, and EPSCs were evoked at every 15 s. AMPAR-mediated EPSCs were recorded at −70 mV, and 20 consecutive responses were recorded after stable baseline. After recording AMPAR-mediated EPSCs, the holding potential was changed to +40 mV to record NMDAR-mediated EPSCs. NMDA component was measured at 60 ms after the stimulation. The NMDA/AMPA ratio was determined by dividing the mean value of 20 NMDAR EPSCs by the mean value of 20 AMPAR EPSC peak amplitudes.

Somatic whole-cell recordings of mEPSCs were obtained in hippocampal CA1 pyramidal neurons at a holding potential of −70 mV. TTX (1 µM) and picrotoxin (100 µM) were added to ACSF to inhibit spontaneous action potential-mediated synaptic currents and IPSCs, respectively. For mIPSCs in hippocampal CA1 pyramidal neurons, recording pipettes (2.5–3.5 MΩ) were filled with an internal solution containing (in mM): 120 CsCl, 10 TEA-Cl, 8 NaCl, 10 HEPES, 5 QX-314-Cl, 4 Mg-ATP, 0.3 Na-GTP, and 10 EGTA, with pH 7.35, 280 mOsm. TTX (1 µM), NBQX (10 µM) and D-AP5 (50 µM) were added to ACSF to inhibit spontaneous action potential-mediated synaptic currents, AMPAR-mediated currents, and NMDAR-mediated currents, respectively. Data were acquired using Clampex 10.2 (Molecular Devices) and analyzed using Clampfit 10 (Molecular Devices). Drugs were purchased from Abcam (TTX), Tocris (NBQX, D-AP5) and Sigma (picrotoxin, DCS).

## Immunoblot analysis

P2 (crude synaptosomes) and PSD I (postsynaptic density I) samples were prepared, as previously described (*Cho et al., 1992*; *Huttner et al., 1983*). CA1 and CA3 regions of the hippocampus were dissected from sagittal slices (300 µm sections). Samples from six slices were pooled and centrifuged at 3000 x g for 1 min, and the pellet was resuspended in homogenization buffer and boiled for 15 min. Antibody used in this papers are followings: PTPσ (home-made, #2135(N-term), #2138(C-term)), PSD-93 (#1634)/SynGAP1 (#1682)/NGL-3 (#2020)/SALM5 (#1943)/GluA1(#1193) (home-made), tubulin (12G10, DSHB), GluN1 (mab363)/GluN2A (07–632) (Millipore), GluN2B (73-101)/PSD-95 (75-028) (neuromab), Shank3 (H160)/Synaptophysin (D4) (Santa Cruz), liprin-α3 (169 102)/caskin 1 (185 003) (Synaptic Systems), liprin-α2 (ab155411)/Trio (194365) (abcam), β-catenin (610154, BD Science)/N cadherin (33–3900, Thermo), TrkC (3376, Cell Signaling), and β-actin (a5316, Sigma).

## Virus preparation and injection

AAV1-hSyn-Cre-eGFP (pENN.AAV.hSyn.HI.eGFP-Cre.WPRE.SV40) and AAV1-hSyn-ΔCre-eGFP (pENN.AAV.hSyn.eGFP.WPRE.bGH) were a gift from James M. Wilson (Addgene #105539-AAV1, #105540-AAV1). WT mouse Ptprs cDNA (*Li et al., 2015*) was subcloned into pAAV-nEFCas9. Point and domain-deleting mutations were introduced to the Ptprs cDNA using overlapping PCR (see *Supplementary file 1* for details). Plasmids containing the shRNA of GluN1 and its scrambled control were purchased (pAV-pU6-shGluN1-GFP; Vigene SH836303, sh#2; 5′-AAGAGAGTGCTGATGTC TTCCAA-3′).

AAVs were prepared using HEK293T cells at 90% confluency and 10 µg of target plasmids (abovementioned), 20 µg of php.eB plasmid (a kind gift from Dr. V. Gradinaru), and 10 µg of pAAV-helper were co-transfected using PEI transfection method (Polysciences, #23966–1). Media collected at 24, 72, and 120 hr after transfection were mixed with 1/5 vol of 40% w/v PEG 8000 (Sigma, #89510) and 2.5 M NaCl solution and centrifuged at 4000 x g for 30 min. Pellets were suspended in

SAN-digestion buffer with 100 U/ml SAN (HL-SAN, Arcticzymes, #70910–202; buffer: 25 mM Tris-HCl, pH 8.5, 5 mM MgCl$_2$, 0.5 M NaCl). The resuspended viruses were centrifuged on iodixanol gradient (Optiprep; Sigma D1556) at 350,000 x g (70Ti ultracentrifuge rotor; 135 min), and the samples at the interface of 42/60% iodixanol were collected using a 16 G needle. The samples diluted in Dulbecco's phosphate-buffered saline (GIBCO) were filtrated using 0.20 µm syringe filter and dialyzed in DPBS using Amicon Ultra-15 (100 kDa cutoff). The virus solution (~150 µL) was stored at −80°C until use in 10 µL aliquots.

For virus injection for *Figure 6*, mice were anesthetized in 1.2% tribromoethanol (20 ml/kg; Sigma; T48402) and placed in a stereotaxic apparatus (Kopf Instruments, Tujunga, CA, USA). The exact injection sites for CA3 and CA1 regions were as follows; CA1, DV = −1.2, ML = ±1.2, AP = −1.94; CA3, DV = −2.09, ML = ±2.3, AP = −1.76. For virus injections for *Figures 2* and *3*, the procedures were identical except for that mice were anesthetized with isofluorane (Piramal Healthcare), and that targeted locations for CA3/CA1 were as follows; CA1, DV = −1.05, ML = ±0.9, AP = −1.9; CA3, DV = −1.9, ML = ±1.7, AP = −1.7. Their bregma-lambda length was 3.6 cm, while the length of adults were 4.2 cm.

qRT-PCR cDNAs were synthesized using TOPscript Cdna synthesis kit (Enzynomics, EZ005). qPCR was performed using SsoAdvanced SYBR Green Supermix (BIORAD, 170-8882AP) and CFX96 Real-Time system. The following primer sets were used in PCR; GAPDH allele, forward, 5'-GTCAGTGG TGGACCTGACCT-3', reverse, 5'-AGGGGAGATTCAGTGTGGTG-3'; GluN1 allele, forward, 5'-AGAGCCCGACCCTAAAAGAA-3', reverse, 5'-CCCTCCTCCCTCTCAATAGC-3'.

## Brain imaging

To examine the gross morphology of the brain, coronal sections (50 µm) of mouse brains were prepared using a vibratome (Leica) and mounted on DAPI-containing Vectashield (Vector Laboratory). For X-gal staining, coronal sections (100 µm) of mouse brains were prepared using a vibratome (Leica) followed by X-gal staining for 30 min (20 mg/mL X-gal; in 2 mM MgCl$_2$, 5 mM K$_4$Fe(CN)$_6$.3H$_2$O(Sigma #P-8131), 5 mM K$_3$Fe(CN)$_6$, 0.01% DOC, 0.02% NP-40 in 1 x PBS). For immunofluorescence imaging of brain sections after electrophysiological and behavioral experiments, coronal brain sections (50 µm) were prepared and used for image acquisition without staining using a confocal microscope (LSM-780, Zeiss).

## Phosphoscan proteomic analysis

Changes in phospho-tyrosine levels in proteins from *Emx1-Cre;Ptprs$^{fl/fl}$* mice were determined using PhosphoScan service (Cell Signaling Technology). Briefly, mouse brain samples containing the cortex and hippocampus were dissected in ice-cold dissection buffer (see the Materials and methods for electrophysiology) with protease/phosphatase inhibitor cocktails. Brain samples from three different mice were pooled to make n number of one. Brain samples were snap-frozen in liquid nitrogen were protease-digested and fractionated by solid-phase extraction. The fractionated peptides were incubated with designated immobilized PTM (post-translational modification)-motif antibodies, and the peptides containing the corresponding PTM-sequences were eluted and analyzed using LC-MS/MS. Mass spectra were assigned to peptide sequences using Socerer program. Finally, the peptide sequence assignment was linked to parent ion peak intensities to measure approximate fold-changes in validated peptides between paired samples.

## Novel object recognition test

Novel object recognition test was performed in the open-field box. On day 1, mice were allowed to explore a novel object (white cylinder) On day 2, mice explored two identical objects (blue cylinder or silver-colored box) for 20 min. On day 3, mice were placed in the same box where one of the two objects was replaced with a new object (blue cylinder and silver-colored box). Sniffing time for each object was measured. Object exploration was defined by the mouse's nose being oriented toward the object and came within 2 cm of the target as measured by EthoVision XT12 program (Noldus).

## Three-chamber test

The three-chambered apparatus, designed to measure social approach and social novelty recognition (*Silverman et al., 2010*), had the dimensions of 40 cm W x 20 cm H x 26 cm D with a center

chamber of 12 cm W and side chambers of 14 cm W. In the first session, the mouse could freely move around the whole three-chambered apparatus with two small containers in the left or right corner for 10 min (Session #1). The mouse was then gently guided to the center chamber while a novel 'Object' and a wild-type stranger mouse 'Stranger 1 (129Sv strain)' were placed in the two plastic containers. The subject mouse was then allowed to freely explore all three chambers for 10 min (Session #2). In the third session, the subject mouse was again gently guided to the center chamber while the 'Object' was replaced with a wild-type 'Stranger 2' mouse. The subject mouse again freely explored all three chambers for 10 min (Session #3). Object/Stranger exploration was defined by the mouse's nose being oriented toward the target and came within 2 cm of it as measured by EthoVision XT 12 program (Noldus). Three-chamber tests over 5-consecutive days were performed as described previously (*Bariselli et al., 2018*). For this experiment, we used mice that did not experience other behavioral tests to minimize potential confounds. The same stranger was exposed to the subject mouse during the first four days. Minor differences in this test, compared with the above mentioned three-chamber test, were the lack of session #3, the use of empty space instead of an object, and 5-min-long session #1 during days 2–5 (10 min for session #1 on day 1).

## Water-based Y-maze

The Y-maze test was performed as described previously (*Trinh et al., 2012*). The Y-maze apparatus was composed of three identical arms (35cm-long, 10cm-wide, 25 cm high). The Y-maze apparatus was placed at the center of a water tank (120 cm diameter) and the platform was placed in one of the three arms and hidden 2 cm under the water (20–22℃) made invisible by white paint. On day 1, a subject mouse was placed in an arm without the hidden platform and allowed to freely swim until it finds the platform. Mice that cannot find the platform in 2 min were guided to the platform. Each session consisted of 5 swim trials, and four sessions were performed on each day. On day 2, a subject mouse was tested for the memory of the platform location for one session. Only the mice that were successful in identifying the correct arm over 80% of the time were used for the following experiments, where the platform location was changed to the opposite arm that was empty on the previous day. The day two experiments consisted of four sessions (five swims per session).

## Morris water maze

Mice were trained to find the hidden platform (10 cm diameter) in a white plastic tank (120 cm diameter). Mice were given three trials per day with an inter-trial interval of 30 min. Experiments for the learning phase of the water maze were performed for seven consecutive days, followed by the probe test on day eight where mice were given 1 min to find the removed platform. For reversal training (days 9–13), the location of the platform was switched to the opposite position from the previously trained location, and mice were allowed to re-learn the new position of the platform. Target quadrant occupancy and the exact number of crossings over the former platform location during the probe test were measured using EthoVision XT12 program (Noldus).

## Laboras test

For long-term measurements of mouse movements, we used the LABORAS system (Metris) (*Quinn et al., 2006*), designed to detect and analyze vibrations delivered from a cage with a mouse to a carbon-fiber vibration-sensitive plate placed underneath the cage. Each mouse was placed in the LABORAS cage without habituation, and its movements were recorded for 72 consecutive hours. The data during the last 48 hr, a period after full habituation to the environment, were analyzed by the software.

## Open-field test

Mice were placed in an open field box (40×40×40 cm) and recorded with a video camera for 60 min. The center zone lines were 10 cm apart from the edge. The testing room was illuminated at ~50 lux or 0 lux. Mice movements were analyzed using EthoVision XT12 program (Noldus).

## Rotarod test

Mice were placed on the rotating rod for 10 s, followed by the start of rod rotation. The rotating speed of rod was gradually increased from 4 to 40 rpm over 5 min. The assay was performed for five

consecutive days, while measuring the latencies of mice falling from the rod or showing 360-degree rotation on the rod.

## Elevated plus-maze test

The elevated plus-maze consisted of two open arms, two closed arms, and a center zone, and was elevated to a height of 50 cm above the floor. Mice were placed in the center zone and allowed to explore the space for 8 min. The data were analyzed using EthoVision XT12 program (Noldus).

## Light-dark test

The light-dark apparatus consisted of light (~200 lux) and dark (~0 lux) chambers adhered to each other. The size of the light chamber was 20×30×20 cm, and that of the dark chamber was 20×13×20 cm. An entrance enabled mice to freely move across the light and dark chambers. Mice were introduced to the center of the light chamber and allowed to explore the apparatus freely for 5 min. The time spent in dark and light chambers and the number of transitions were measured using EthoVision XT12 program (Noldus).

## Prepulse inhibition

A subject mouse was placed in a startle chamber (SR-LAB). For acclimation, a background noise of 65 dB pulse was given for 5 min. After acclimation, 57 testing sound pulses with varying inter-trial intervals (7–23 s) were given. The testing sound pulses consist of 4 pulses (4 × 120 ms, 120 dB) in the beginning and end stage of the test, and seven pulses (120 ms, 120 dB each) paired with pre-pulses (20 ms 100 ms prior to) at 70, 75, 80, 85 and 90 dB (total 35 paired pulses).

## Statistics

For statistical comparison of two samples (e.g., WT vs. cKO), Student's t-test or Mann-Whitney test was used. The normality of data distributions was tested using the D'Agostino and Pearson normality test or Shapiro-Wilk normality test. Mann-Whitney tests were used for any column in either of the two tests in which the p-value was less than 0.05. For immunoblot and qRT-PCR results, a one-sample t-test was used. For results with one independent variable [e.g., cKO-eGFP vs. cKO-Ptprs(4A) vs. cKO-Ptprs(Y224S)], one-way analysis of variance (ANOVA) with Dunnett's multiple comparison test was used. For results with two independent variables (e.g., WT-Veh vs. WT-DCS vs. cKO-Veh vs. cKO-DCS), two-way ANOVA with Sidaks' multiple comparison test was used. For additional information on gender and number of mice/samples, detailed test information and statistical results, see *Supplementary file 3*. GraphPad Prism seven was used for all statistical analyses, except for the DAVID analysis (http://david.ncifcrf.gov).

## Acknowledgements

We would like to thank Drs. Peter Scheiffele and Dietmar Schreiner at the University of Basel for helpful comments on the manuscript. This work was supported by the National Research Foundation of Korea (NRF) grant funded by the Korean government (MSIT, NRF-2017R1A5A2015391 to YCB) and the Institute for Basic Science (IBS-R002-D1 to EK).

## Additional information

### Competing interests

Eunjoon Kim: Reviewing editor, *eLife*. The other authors declare that no competing interests exist.

### Funding

| Funder | Grant reference number | Author |
| --- | --- | --- |
| Institute for Basic Science | IBS-R002-D1 | Eunjoon Kim |
| National Research Foundation of Korea | MSIT, NRF-2017R1A5A2015391 | Yong Chul Bae |

The funders had no role in study design, data collection and interpretation, or the decision to submit the work for publication.

### Author contributions
Kyungdeok Kim, Wangyong Shin, Muwon Kang, Suho Lee, Ryeonghwa Kang, Data curation, Investigation; Doyoun Kim, Investigation, Visualization; Yewon Jung, Yisul Cho, Esther Yang, Data curation; Hyun Kim, Yong Chul Bae, Supervision; Eunjoon Kim, Supervision, Funding acquisition, Investigation, Writing - original draft, Project administration, Writing - review and editing

### Author ORCIDs
Kyungdeok Kim (ID) https://orcid.org/0000-0002-0003-6957
Eunjoon Kim (ID) https://orcid.org/0000-0001-5518-6584

### Ethics
Animal experimentation: All mice were housed and bred at the mouse facility of Korea Advanced Institute of Science and Technology (KAIST) and maintained according to the Animal Research Requirements of KAIST. All procedures were approved by the Committee of Animal Research at KAIST (KA2016-33).

### Decision letter and Author response
Decision letter https://doi.org/10.7554/eLife.54224.sa1
Author response https://doi.org/10.7554/eLife.54224.sa2

## Additional files

### Supplementary files
• Supplementary file 1. Amino acid sequences of PTPσ mutants used in the present study.
• Supplementary file 2. Details of the results of pTyr analysis.
• Supplementary file 3. Statistical details.
• Transparent reporting form

### Data availability
All data generated or analysed during this study are included in the manuscript and supporting files.

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
