## [Decision Letter]

**Acceptance summary:**

As nicely summarized by one of the reviewers, the authors investigated the role of the presynaptic protein PTPσ, a LAR-type receptor phosphotyrosine-phosphatase, as a transsynpatic regulator of postsynaptic NMDA receptors. They developed a conditional KO mouse (PTPσ-cKO) and provide evidence that deletion of PTPσ reduces postsynaptic NMDAR-mediated transmission and NMDAR-mediated plasticity (LTP and LTD) in the hippocampus that PTPσ-dependent regulation of NMDARs likely involves interactions of PTPσ with cytoplasmic proteins that presumably link PTPσ to phosphotyrosine substrates; and that deletion of PTPσ is associated with impairments in social and reward-related novelty recognition. The findings support the notion that a presynaptic adhesion protein can control postsynaptic function in a non-canonical way that does not involve trans-synaptic protein-protein interactions.

**Decision letter after peer review:**

Thank you for submitting your article "Presynaptic PTPσ Regulates Postsynaptic NMDA Receptor Function through Cytoplasmic Domains" for consideration by *eLife*. Your article has been reviewed by Gary Westbrook as the Senior Editor, a Reviewing Editor, and three reviewers. The following individuals involved in review of your submission have agreed to reveal their identity: Pablo E Castillo (Reviewer #1); Johannes W Hell (Reviewer #2); Matthijs Verhage (Reviewer #3). The reviewers have discussed the reviews with one another and the Reviewing Editor has drafted this decision to help you prepare a revised submission.

Summary:

All three reviewers were interested in the topic and supportive of the work. However, they raised several issues, most of which we think can be dealt with by adjustments in the text and Discussion section. These are enumerated in the comments of the reviewers below.

*Reviewer #1:*

In this thorough study, the authors investigated the role of the presynaptic protein PTPσ, a LAR-type receptor phosphotyrosine-phosphatase, as a trans-synpatic regulator of postsynaptic NMDA receptors. To this end, they developed a conditional KO mouse (PTPσ-cKO) and used multiple approaches, including synaptic electrophysiology in hippocampal slices, proteomics and behavioral tests. The authors provide experimental evidence in support of the following three main findings. First, deletion of PTPσ reduces postsynaptic NMDAR-mediated transmission and NMDAR-mediated forms of plasticity (LTP and LTD) in the hippocampus. Second, PTPσ-dependent regulation of NMDARs likely involves interactions of PTPσ with cytoplasmic proteins that presumably link PTPσ to phosphotyrosine substrates. Third, deletion of PTPσ is associated with significant impairments in social and reward-related novelty recognition. The findings are exciting and support the notion that a presynaptic adhesion protein can control postsynaptic function in a non-canonical way that does not involve trans-synaptic protein-protein interactions. The study includes a large number of well-designed experiments. The results are clean and support most of the claims that are made. The authors' findings will be of interest to a large audience of neuroscientists and cellular biologists.

Essential revisions:

The authors claim that "PTPσ trans-synaptically regulates the postsynaptic localization and function of NMDARs in the hippocampus". However, their study provides rather indirect evidence that PTPσ changes the localization of NMDARs. The results presented in Figure 1J suggest a global reduction in GluN2B subunits. The authors may want to tone down their claim and provide some explanation for this global change.

Figure 6A: The starting point (familiar) seems to be significantly different between WT and PTPσ-cKO mice. Why? How would this difference affect the interpretation of the results?

The authors indicate that PTPσ is expressed in the cerebellum but the images suggest otherwise (Figure 1—figure supplement 1G).

What is the authors' explanation for a reduced LTD in PTPσ-cKOs?

Did D-cycloserine rescued NMDAR-mediated transmission (e.g. NMDAR-EPSCs) in PTP-cKO mice?

Discussion section. Indicate whether or not previous studies have specifically linked postsynaptic hippocampal LTP/LTD to novel recognition tasks.

Validating the knockdown in cultures is okay but not ideal. The authors may want to show the efficacy of the knockdown in tissue.

"Evoked EPSCs at Schaffer collateral-CA1 pyramidal (SC-CA1) synapses…", add the word "cell" after pyramidal.

*Reviewer #2:*

PTPs may contribute to adhesion between pre-and postsynaptic sites but numerous aspects of the roles of PTPs in synapse formation and function remain unclear. In this manuscript Eunjoon Kim and his co-workers create conditional KO mice in which PTPσ was deleted in excitatory neurons in cortex and hippocampus. In these mice basal synaptic transmission at the CA3-CA1 synapse was normal including mEPSC amplitude and frequency, evoked transmission and PPR. These findings contrast earlier work in global PTPs KO mice. However, the authors find that LTP as well as LTD were impaired in these cKO. Loss of LTP was rescued by boosting NMDAR activity with D-cycloserine. These data suggest that in these cKO mice postsynaptic NMDAR activity is reduced. Consistent with these functional studies, the content of the NMDAR GluN2B subunit was decreased in PSD fractions.

To define whether pre-or postsynaptic PTPσ expression is relevant here, the authors deleted PTPσ by viral injection in either CA3 or CA1 neurons. Only presynaptic (CA3) but not postsynaptic (CA1) deletion impaired LTP. Using the cKO mice in which PTPσ was deleted in excitatory forebrain neurons the authors further demonstrate that re-expression of WT PTPσ in presynaptic CA3 neurons rescued the LTP. This was also true for expression of several different PTPσ constructs with point and deletion mutations that impaired interactions with different postsynaptic binding partners including HSPG/CSPG and Slitrk/TrcC. However, expression of PTPσ with a point mutation that impaired phosphatase activity and two deletion of intracellular regions that impaired binding to caskin were not able to rescue LTP.

To gain further molecular insight the authors performed an unbiased proteomic approach to define changes in tyrosine phosphorylation in the cKO mice in which PTPσ was deleted in excitatory forebrain neurons. Accordingly, ~1500 proteins showed changes and many of those are synaptic proteins. Presynaptic proteins showed more increases in pTyr as can be expected upon loss of a phosphatase whereas postsynaptic proteins showed more decreases in pTyr. The latter includes several tyrosine residues in GluN2B, which could potentially explain the reduction of GluN2B content of the PSD and thereby reduction in NMDAR activity, as described above.

Finally, the authors performed a number of learning and memory tasks. In mice in which PTPσ was deleted in excitatory forebrain neurons most initial learning was normal when novelty recognition was impaired in several tests (novel object, novel social mouse, novel reward arm in Y maze, and reversal learning in the Morris Water Maze). Remarkably, preference for novel mouse in the social assay was rescued by systemic administration of D-cycloserine, which augments NMDAR activity. Furthermore, deletion of PTPσ in CA3 but not CA1 neurons impaired the novel mouse preference in the social interaction test as well as novel arm preference in the Y maze. Knockout of GluN1 in the postsynaptic CA1 neurons phenocopied these two effects. Collectively these results indicate that loss of presynaptic PTPσ affects postsynaptic NMDAR function and thereby novelty learning.

The above findings are of very high functional significance. This is a very rigorous, comprehensive, and thorough study with functional effects on multiple levels including NMDAR activity in synaptic transmission and novelty learning.

In Figure 6—figure supplement Figure 1 data suggest decreased baseline anxiety including a tendency to increased open field time and a significant increase in open arm time in the elevated plus maze-that should be more detailed in the Results section and discussed.

*Reviewer #3:*

This study describes how conditional deletion of PTPσ alters synaptic plasticity and posttranslational modification of proteins on either side of the synapse and attempts to connect these alterations to altered cognitive bahaviors. The manuscript presents a massive amount of data, an important new mouse model and important new conclusions on PTPσ function and the trans-synaptic regulation of synaptic strength. These conclusions are different from previous conclusions obtained using classical PTPσ-null mice and challenge the existing concept of how PTPs regulate synapse differentiation and plasticity. The novel concept, postulating that PTP acts through presynaptic interactions with other (trans-synaptic) molecules to alter postsynaptic receptor function and balance, is new, important and convincing. The conclusions are built on an impressive series of in vivo rescue experiments and state of the art phospho-proteomics and analysis. The SynGO analysis of altered pTyr levels presents a spectacular difference between pre- and postsynaptic pTyr proteins (with presynaptic proteins all having increased pTyr levels and postsynaptic proteins decreased pTyr levels). The altered pTyr proteins include relevant new substrates on either side of the synapse.

Essential revisions:

1) PTPσ mutant constructs carrying mutations/deletions in cytoplasmic domains D1-2 do not rescue LTP, which is a central element in the argumentation, but it is unknown if these constructs are targeted correctly. Some evidence for correct targeting is required. Most convincing and more positive evidence would be to show that targeting of the cytoplasmic domains (e.g. using GAP43) is sufficient for normal LTP, but this is probably beyond the scope of the current study.

2) While it seems easy to explain why presynaptic proteins have almost exclusively higher pTyr levels, it is quite intriguing why postsynaptic proteins almost exclusively have lower pTyr levels. Given the proposed indirectness of these effects and the fact that so many postsynaptic substrates are altered, it is quite remarkable that the postsynaptic effects are so one-directional. This should be discussed.

3) Behavioral analyses are extensive and interesting but the connection to cellular/slice data remains loose. Although the behavioral abnormalities are certainly consistent with altered hippocampal NMDA function, the behavioral effects are most likely the net result of many (unknown) cellular and network effects of Emx-expressing neurons, their targets and the targets of these targets etc. A more balanced discussion on this topic would be preferable.

---

## [Author Response]

Reviewer #1:Essential revisions:The authors claim that "PTPσ trans-synaptically regulates the postsynaptic localization and function of NMDARs in the hippocampus". However, their study provides rather indirect evidence that PTPσ changes the localization of NMDARs. The results presented in Figure 1J suggest a global reduction in GluN2B subunits. The authors may want to tone down their claim and provide some explanation for this global change.

We agree with the indirect nature of the regulation and changed the last sentence in the Abstract as follows: “These results suggest that presynaptic PTPσ regulates postsynaptic NMDAR function and novelty recognition in social and reward contexts.” was changed to “These results suggest that presynaptic PTPσ regulates postsynaptic NMDAR function through trans-synaptic and direct adhesion-independent mechanisms and novelty recognition in social and reward contexts.”.

Regarding the reduction of GluN2B levels in Figure 1J, the results shown are from P2 (crude synaptosomal) and PSD fractions; we apologize for the misleading. When total lysates of the hippocampus are used, there is no genotype difference (added to Figure 1J), suggesting that synaptic localization, but not total levels, of GluN2B is decreased.

Figure 6A: The starting point (familiar) seems to be significantly different between WT and PTPσ-cKO mice. Why? How would this difference affect the interpretation of the results?

The increase in baseline exploration (~2-folds) in PTPσ-cKO mice might be partly attributable to that these mice are ~20% more locomotive and ~30% more active in exploring the objects (n = 16 mice), compared with WT mice. However, we doubt that these factors would affect the ‘relative’ exploration of familiar and novel objects. We commented on this in Results section.

The authors indicate that PTPσ is expressed in the cerebellum but the images suggest otherwise (Figure 1—figure supplement 1G).

We agree with the reviewer and removed ‘cerebellum’ from the text.

What is the authors' explanation for a reduced LTD in PTPσ-cKOs?

We suspect a decrease in NMDAR currents rather than signaling pathways downstream of NMDAR activation, based on the following text newly added to Results section: “In addition, considering the extents of the decreases in HFS-LTP, TBS-LTP, and LFS-LTD (~44%, ~66%, and ~53%, respectively) and the decrease in the NMDA/AMPA ratio (~45%) at the mutant synapses under naïve and D-cycloserine rescue conditions (Figure 1E-I and 1K,L), the decreased LTP and LTD seem to mainly involve decreased NMDAR currents rather than signaling pathways downstream of NMDAR activation. In addition, the decreased levels of GluN2B in the PSD fraction (~20%) may contribute partly to the decrease in NMDAR currents (~45%).”

Did D-cycloserine rescued NMDAR-mediated transmission (e.g. NMDAR-EPSCs) in PTP -cKO mice?

We performed the suggested experiment and found that the NMDA/AMPA ratio was rescued by D-cycloserine (Figure 1K), suggesting that D-cycloserine rescues LTP through the normalization of NMDAR currents.

Discussion section. Indicate whether or not previous studies have specifically linked postsynaptic hippocampal LTP/LTD to novel recognition tasks.

We modified Discussion section as follows: “In addition, hippocampal LTD but not LTP has been associated with novel-object/feature recognition in a space (Dong et al., 2012; Kemp and Manahan-Vaughan, 2004, 2007). […] Whether our results would also involve the rescue of NMDAR-dependent LTD, which is impaired in the *Emx1-Cre;Ptprs^fl/fl^* hippocampus, remain to be determined.”

Validating the knockdown in cultures is okay but not ideal. The authors may want to show the efficacy of the knockdown in tissue.

In response, we validated GluN1 knockdown in the hippocampus by the immunoblot analysis of the GluN1 protein (Figure 6H).

"Evoked EPSCs at Schaffer collateral-CA1 pyramidal (SC-CA1) synapses…", add the word "cell" after pyramidal.

Corrected. Again, we appreciate the very helpful comments of the reviewer!

Reviewer #2:In Figure 6—figure supplement 1 data suggest decreased baseline anxiety including a tendency to increased open field time and a significant increase in open arm time in the elevated plus maze-that should be more detailed in the Results section and discussed.

We detailed these behavioral changes in Results section and made relevant comments in Discussion section.

Reviewer #3:Essential revisions:1) PTPσ mutant constructs carrying mutations/deletions in cytoplasmic domains D1-2 do not rescue LTP, which is a central element in the argumentation, but it is unknown if these constructs are targeted correctly. Some evidence for correct targeting is required. Most convincing and more positive evidence would be to show that targeting of the cytoplasmic domains (e.g. using GAP43) is sufficient for normal LTP, but this is probably beyond the scope of the current study.

We appreciate this comment. In response, we performed presynaptic targeting experiments using cultured hippocampal neurons. To this end, we first considered making GAP43 constructs containing the cytoplasmic domain of WT and mutant PTPσ proteins (C1146S [phosphatase-dead], ΔD2 [D2 domain deletion], and EGFID-to-AGFAA [loss of caskin binding]). However, we reasoned that the removal of the ectodomain in the PTPσ protein, well-known to trans-synaptically interacts with various postsynaptic adhesion molecules (i.e. NGL-3, SALMs, TrkC, IL1RAcP, and Slitrks), would greatly reduce their presynaptic localization, assuming that both ectodomain and cytoplasmic domain interactions would fully stabilize PTPσ at presynaptic sites. We thus attempted HA tagging at the C-terminus of PTPσ (WT and mutants) and tested their presynaptic targeting in cultured neurons, although this tag may also interfere with the cytoplasmic interactions.

We found that both WT and mutant PTPσ proteins are similarly targeted to presynaptic boutons, but only a small fraction of them were in contact with vGluT1-positive Shank2 puncta, marking excitatory synaptic sites. This suggests, first, the possibility that both WT and mutant proteins are equally targeted to presynaptic sites, without being affected by the mutations, and supports our hypothesis that the cytoplasmic domains/motifs are indeed important for the PTPσ-dependent LTP regulation. Alternatively, this result could mean that WT and mutant PTPσ proteins with an HA tag coupled to the protein C-terminus cannot be properly targeted to presynaptic nerve terminals by some negative influences of the HA tag on both WT and mutant proteins.

Because a careful and comprehensive validation of these mutant PTPσ proteins for their synaptic targeting and protein-protein interactions would take a significant amount of time and effort, and it is important to publish only solid sets of data, and this dataset on the cytoplasmic mutations comprises only a small portion of the whole dataset, we decided to remove Figure 4B (the results related with the cytoplasmic mutations) from the revised manuscript. The manuscript still retains the main conclusions intact; presynaptic PTPσ regulates postsynaptic NMDAR currents and LTP through trans-synaptic and adhesion-independent mechanisms and is important for various forms of novelty recognition in mice. Accordingly, we made relevant changes in the text and figures as follows; (1) Figure 4B was removed, (2) A part of the diagrams and immunoblot data in Figure 3B-D were modified (to remove cytoplasmic domain-related diagram/data), (3) Figure 3 and Figure 4 (without panel B) were merged into Figure 3, and (4) relevant texts were modified throughout the manuscript, including the Title and Abstract. The remaining mechanisms related with the role of the cytoplasmic domains of PTPσ in NMDAR regulation will be explored in future studies.

2) While it seems easy to explain why presynaptic proteins have almost exclusively higher pTyr levels, it is quite intriguing why postsynaptic proteins almost exclusively have lower pTyr levels. Given the proposed indirectness of these effects and the fact that so many postsynaptic substrates are altered, it is quite remarkable that the postsynaptic effects are so one-directional. This should be discussed.

This is an interesting point but has to be speculated at the moment. We made the following comments in Discussion section: “Mechanisms underlying the strong postsynaptic decreases (not increases) in pTyr levels are unclear but may involve altered activities of tyrosine kinases and phosphatases at postsynaptic sites, as suggested by altered p-Tyr levels in these proteins (Supplementary file 1), or the decreased function of NMDARs, known to bidirectionally interact with and regulate various tyrosine kinases and phosphatases such as Src family proteins, Eph receptors, and STEP (Henderson and Dalva, 2018; Sala and Sheng, 1999; Salter and Kalia, 2004).”.

3) Behavioral analyses are extensive and interesting but the connection to cellular/slice data remains loose. Although the behavioral abnormalities are certainly consistent with altered hippocampal NMDA function, the behavioral effects are most likely the net result of many (unknown) cellular and network effects of Emx-expressing neurons, their targets and the targets of these targets etc. A more balanced discussion on this topic would be preferable.

We agree with the reviewer and added the following comments in Discussion section: “Care should be taken, however, in interpreting our results because *Emx1-Cre;Ptprs^fl/fl^* mice lack *Ptprs* expression not only in the hippocampus but also in other brain regions such as the prefrontal cortex, known to be involved in social novelty cognition in mice and rats (Cao et al., 2018; Finlay et al., 2015; Liang et al., 2018; Niu et al., 2018; Watson et al., 2012)”.